# Counterfactual Prediction with Cross-World Dependence

## Abstract

We study the problem of estimating the expected retrospective counterfactual outcome for an individual with covariates $x$ and observed outcome $y$, defined as $\mu(x, y) = \mathbb{E}[Y(1) \mid X = x, Y(0) = y]$, and constructing valid prediction intervals under the Neyman–Rubin superpopulation model with i.i.d. units. This quantity is generally unidentified without additional assumptions. To link the observed and unobserved potential outcomes, we work with a cross-world correlation function $\rho(x) = \text{cor}(Y(1), Y(0) \mid X = x)$ that quantifies their dependence given the covariates. Plausible bounds on $\rho(x)$, often informed by domain knowledge, enable a principled approach to this otherwise unidentified problem. Given $\rho$, we develop an estimator $\hat{\mu}_\rho(x, y)$ and prediction intervals $C_\rho(x, y)$ that satisfy $P[Y(1) \in C_\rho(X, Y(0))] \geq 1 - \alpha$ under standard causal assumptions and Gaussian dependence structure. Almost all existing methods correspond to either the case $\rho = 0$ (ignoring the factual outcome), or $\rho = 1$ (constant treatment effects). We show that interpolating between these cases via cross-world dependence yields estimators that are theoretically optimal under (asymptotic) Gaussian assumptions. In practice, this leads to substantial empirical improvements across a wide range of scenarios.

## 1 Introduction

At its core, causal inference pursues two goals: assessing what would have happened to an individual under an alternative treatment, and predicting how a new individual will benefit from treatment (Rubin, 2005). For answering the second goal, the literature focuses on average treatment effects (ATE) or conditional average treatment effects (CATE). However, estimating retrospective counterfactuals (first goal) is often more challenging, as it requires untestable assumptions, connected to the Pearl's third ladder of causation (Pearl & Mackenzie, 2019). Estimates of counterfactuals are critical in many fields: in medicine, they enable evaluating how a patient might have responded to a different treatment (Imbens & Rubin, 2015); in criminal law, they underpin the "but-for" test of causation, which assesses liability based on whether harm would have occurred absent the defendant's action (Wright, 1985).

Consider a medical scenario in which a patient, James, arrives at a hospital with covariates $X = x$ (e.g., age, weight, and other characteristics), does not receive the treatment ($T = 0$), and experiences an outcome $Y(0) \in \mathbb{R}$. Estimating his retrospective counterfactual outcome $Y(1)$ is central to causal reasoning. In high-stakes settings such as healthcare, it is equally important to quantify the uncertainty in individual treatment effects (ITEs); that is, to construct a set $C \subseteq \mathbb{R}$ that contains $Y(1)$ with high probability.

Existing methods primarily focus on estimating the CATE, defined as $\tau(x) = \mu_1(x) - \mu_0(x)$, where $\mu_t(x) = \mathbb{E}[Y(t) \mid X = x]$ for $t = 0, 1$ can be estimated via e.g. random forest (Wager & Athey, 2018). The missing counterfactual is often imputed either by $\hat{Y}(1) = Y(0) + \hat{\tau}(X)$, by $\hat{Y}(1) = \hat{\mu}_1(X)$, or through a matching-based approach. Some notable exceptions are presented in Section 2 and Appendix A.1.

Many existing approaches condition only on covariates $X$, overlooking the observed (factual) outcome $Y(0)$, which often contains valuable individual-specific information. For instance, if James left the hospital healthy after not receiving treatment ($T = 0$), it is highly likely that he would also

be healthy under the counterfactual scenario in which he received treatment ($T = 1$). Incorporating the factual outcome alongside the covariates can therefore refine individual-level predictions and improve the accuracy of estimated counterfactuals.

In this work, we propose leveraging covariates *and* the factual outcome to enhance counterfactual prediction. Specifically, instead of estimating $\mathbb{E}[Y(1) \mid X = x]$, we aim to construct point estimates

$$\hat{\mu}_\rho(x, y) \quad \text{for} \quad \mathbb{E}\left[Y(1) \mid X = x, Y(0) = y\right], \tag{1}$$

and $(1 - \alpha)$-level prediction intervals $C_\rho(x, y)$ for the counterfactuals satisfying:

$$P\left(Y(1) \in C_\rho(x, y) \mid X = x, Y(0) = y\right) \geq 1 - \alpha, \tag{2}$$

for $\alpha \in (0, 1)$ (typically $\alpha = 0.1$). Conditioning on the factual outcome introduces a fundamental challenge: since both potential outcomes are never observed for the same individual, the joint distribution of $\left(Y(0), Y(1)\right)$ is unidentifiable without further assumptions. To address this, we adopt a class of assumptions known as cross-world assumptions.

**Definition 1** (Bodik et al. (2025)). *In the Neyman–Rubin super-population model with i.i.d. units, the dependence structure (conditional correlation) between the potential outcomes $Y(1), Y(0)$, conditioned on the observed covariates $X$, is defined as:*

$$\rho(x) = \text{cor}\left(Y(1), Y(0) \mid X = x\right).$$

*We refer to an assumption about $\rho$ as cross-world assumption.*

The term "cross-world assumption" was first introduced in Bodik et al. (2025), and related ideas have appeared in prior literature (see Section 2), often represented via an additive structural equation model:

$$Y(0) = \mu_0(X) + \varepsilon_0, \quad Y(1) = \mu_1(X) + \varepsilon_1, \quad \text{where} \ \text{cor}(\varepsilon_1, \varepsilon_0) = \rho(X).$$

Although $\rho$ is not identifiable from data, postulating plausible values or bounds from domain experts is often both feasible and well-aligned with how humans make judgments. Observing one potential outcome often conveys information about the other, beyond what is captured by covariates.

**Our contributions.** Given a specified value (or a set of plausible values) of $\rho$, we propose a consistent counterfactual point estimator equation 1 and valid prediction intervals equation 2, under standard causal assumptions and Gaussian copula. For clarity, we focus on the case $T = 0$ and the counterfactual outcome is $Y(1)$; the reverse case is analogous. While the formal definitions of $\hat{\mu}_\rho(x, y)$ and $C_\rho(x, y)$ are given in Section 3, we present here the key property that motivates their construction:

**Theorem 1** (Motivation and optimality). *Let $x \in \mathcal{X}$ and $\rho(x) = \text{cor}\left(Y(0), Y(1) \mid X = x\right) \in [-1, 1]$. Assume an asymptotic scenario: $\hat{\mu}_t(x) = \mu_t(x)$ and suppose that we found conditionally valid prediction intervals:*

$$\mathbb{P}\left(Y(t) \leq \hat{\mu}_t(x) + u_t(x) \mid X = x\right) = 0.95, \quad \mathbb{P}\left(Y(t) \geq \hat{\mu}_t(x) - l_t(x) \mid X = x\right) = 0.95, \quad t = 0, 1.$$

*If $\left(Y(1), Y(0)\right) \mid X = x$ is Gaussian, then $C_\rho$ prediction intervals, defined in Section 3, are optimal in a sense that it is the smallest set satisfying:*

$$\mathbb{P}\left(Y(1) \in C_\rho(X, Y(0)) \mid X = x, Y(0) = y\right) \geq 0.9.$$

*Moreover, $\hat{\mu}_\rho(x, y)$ is the optimal point predictor in the sense that it minimizes the mean squared error:*

$$\hat{\mu}_\rho(x, y) = \underset{c \in \mathbb{R}}{\text{argmin}} \ \mathbb{E}\left[(Y(1) - c)^2 \mid X = x, Y(0) = y\right].$$

Our proposed $C_\rho$ intervals are introduced in Section 3, following preliminaries in Section 2. In Section 4, we discuss empirical evaluation compared to other methods. Section 5 concludes.

## 2 PRELIMINARIES, RELATED WORK AND CROSS-WORLD ASSUMPTION

We adopt the Neyman-Rubin potential outcomes framework (Rubin, 2005), where each unit $i$ has potential outcomes $Y_i(1)$ and $Y_i(0)$, covariates $X_i \in \mathcal{X} \subseteq \mathbb{R}^d$, and treatment assignment $T_i \in \{0,1\}$. The observed outcome is $Y_i = T_i Y_i(1) + (1 - T_i) Y_i(0) \in \mathcal{Y} \subseteq \mathbb{R}$, while the $ITE_i = Y_i(1) - Y_i(0)$ remains unobservable. We assume $(Y_i(1), Y_i(0), T_i, X_i) \overset{\text{i.i.d.}}{\sim} (Y(1), Y(0), T, X)$, for a generic random vector $(Y(1), Y(0), T, X)$. The conditional average treatment effect (CATE) is defined as $\tau(x) = \mu_1(x) - \mu_0(x)$ with $\mu_t(x) = \mathbb{E}[Y(t) \mid X = x]$.

We impose **strong ignorability** and **overlap**, meaning $(Y(1), Y(0)) \perp\!\!\!\perp T \mid X$ and $0 < \pi(x) < 1$ for all $x \in \mathcal{X}$, where $\pi(x) = \mathbb{P}(T = 1 \mid X = x)$ denotes the propensity score. These conditions ensure that treatment is as-if randomly assigned given covariates and that both treatments are feasible. Under these assumptions, CATE is identified via $\mu_t(x) = \mathbb{E}[Y \mid T = t, X = x]$(Wager, 2024).

We note that some authors use the terms "ITE" and "CATE" interchangeably, which can lead to confusion. Here, ITE is a latent, unit-specific quantity, while the CATE is an unknown function, defined as the conditional expectation of the ITE given covariates.

### 2.1 RELATED WORK: CROSS-WORLD ASSUMPTION

In the potential outcomes framework, the joint distribution of $\big(Y(1), Y(0)\big) \mid X$ is unidentifiable because only one potential outcome is observed per unit. While CATE can be identified without assumptions on this joint law, quantities such as variance, quantiles, or prediction intervals of ITE generally depend on the cross-world correlation $\rho(X) = \text{cor}\big(Y(1), Y(0) \mid X\big)$ (Rubin, 1990; Ding et al., 2019). This has been studied in joint distribution modeling (Heckman et al., 1997; Fan & Park, 2010), quantile treatment effect estimation (Firpo, 2007) and nonparametric bounds using copulas (Zhang & Richardson, 2025a;b; Nelsen et al., 2001). Andrews & Didelez (2021) highlight the implausibility of cross-world independence assumptions in mediation analysis; we complement these by parameterizing cross-world dependence via $\rho(x)$.

Bodik et al. (2025) and Cai et al. (2022) argue that in many real-world applications $\rho$ is almost always non-negative and often substantially positive due to shared latent factors affecting both potential outcomes. Formally, consider a model where $Y(1) = \mu_1(X) + H + \tilde{\varepsilon}_1$ and $Y(0) = \mu_0(X) + H + \tilde{\varepsilon}_0$, where $X \in \mathbb{R}^d$ are observed covariates, $H \perp\!\!\!\perp (X, T)$ is an unobserved factor influencing both potential outcomes, and $\tilde{\varepsilon}_0 \perp\!\!\!\perp \tilde{\varepsilon}_1$ are idiosyncratic noise terms. Conditioning on $X$, it is easy to derive that $\rho(X) = \text{cor}(Y(1), Y(0) \mid X) = \frac{\text{var}(H)}{\sqrt{\text{var}(\tilde{\varepsilon}_0) \text{var}(\tilde{\varepsilon}_1)}} \geq 0$. Whenever $\text{var}(H) > 0$, the shared influence of $H$ induces strictly positive correlation between $Y(1)$ and $Y(0)$, even after adjusting for $X$. Moreover, if the treatment has no or very small effect, then $Y(1) \approx Y(0)$ and hence $\rho \approx 1$.

Following Bodik et al. (2025), the choice of $\rho(x)$ can be guided by practitioners by asking: "What proportion of the outcome variability is driven by latent factors that influence both potential outcomes in a similar way?" In other words, what values are plausible for $\frac{\text{var(shared latent effects)}}{\text{var(idiosyncratic noise)}}$. In many complex systems, it is reasonable to expect a substantial contribution from shared latent components, suggesting that $\rho(x)$ may typically exceed $0.5$. At the same time, $\rho(x)$ is rarely close to 1, since treatment effects generally exhibit heterogeneity even among individuals with the same observed covariates $X$. This is not a universal rule, but a practical guideline grounded in the idea how latent common causes in many real-world systems influence both $Y(0)$ and $Y(1)$.

As an example, consider a clinical trial testing a new drug for reducing blood pressure, where the treatment is randomly assigned and standard causal assumptions hold. Let $Y_i(1)$ denote patient $i$'s blood pressure after receiving the drug and $Y_i(0)$ after receiving a placebo. Even though baseline covariates such as age, weight, and existing conditions are observed, unmeasured factors like genetic predisposition can strongly influence both potential outcomes. A patient with naturally resilient cardiovascular health will likely exhibit relatively low blood pressure regardless of treatment, whereas a patient with severe underlying issues will tend to have higher readings in both scenarios. These persistent latent traits induce a positive dependence between $Y_i(1)$ and $Y_i(0)$ even after adjusting for observed covariates. Given this medical knowledge, it is reasonable to assume $\rho(x)$ is not only

positive but possibly large, likely above 0.5. See Bodik et al. (2025) for more examples when some domain knowledge about $\rho$ is available.

## 2.2 Related work: retrospective counterfactuals for in-study units

Inferring individual counterfactual outcomes is fundamentally a missing data problem (Ding & Li, 2018). Many methods for counterfactual prediction use CATE-adjusted imputation $\hat{Y}_i(1) = Y_i(0) + \hat{\tau}(X_i)$, where $\hat{\tau}$ is estimated using doubly-robust estimator, random forests or S/T-learner (Wager, 2024; Künzel et al., 2019; Athey et al., 2019). Other approaches directly model the treated outcome as $\hat{Y}_i(1) = \hat{\mu}_1(X_i)$, thereby ignoring information contained in the observed outcome $Y_i(0)$ (possibly using control group only for the propensity estimation, Lei & Candès (2021)).

Classic counterfactual prediction methods target $\mathbb{E}[Y(T) \mid X]$ without conditioning on $Y(0)$. For instance, Kim et al. (2022) propose a doubly robust estimator for counterfactual classification that directly models the treated outcome distribution, and McClean et al. (2024) develop nonparametric estimators for conditional incremental effects (based on stochastic propensity interventions) with a similar goal of directly estimating $\mathbb{E}[Y(1) \mid X]$. More recently, Kim (2025) introduces a semi-parametric counterfactual regression framework that likewise estimates $\mathbb{E}[Y(1) \mid X]$ using flexible machine learning. These approaches forego individual-level imputation using $Y(0)$, instead relying on robust modeling of the treated outcome. Most existing methods focus on minimizing the Precision in Estimation of Heterogeneous Effects (PEHE), defined as $\mathbb{E}_X\big(\hat{\tau}(X) - \tau(X)\big)^2$, which targets CATE recovery. However, optimizing PEHE is not well suited for inference about counterfactuals.

There are a few notable exceptions where the construction of $\hat{Y}_i(1)$ follows a different principle. **Adversarial approaches**: Yoon et al. (2018) introduce GANITE, which employs adversarial training to generate $\hat{Y}_i(1)$. Although GANITE innovatively bypasses strict model assumptions, it focuses on PEHE and relies on black-box adversarial neural networks without explicitly modeling the joint distribution of potential outcomes. It typically performs well with large dimensions but poorly with small ones. **Bayesian causal inference**: Missing counterfactuals are treated as latent variables, and uncertainty is integrated through the posterior distribution. For example, Alaa & van der Schaar (2017) propose a Bayesian multitask Gaussian process to jointly model $\big(Y(1), Y(0)\big) \mid X$, producing posterior distributions over the potential outcomes. While Bayesian methods offer coherent uncertainty quantification, they rely on strong modeling assumptions and can be sensitive to prior specifications (Li et al., 2022). Moreover, they can be restrictive when aiming to leverage flexible modern machine learning techniques. **Matching methods**: Matching-based approaches (Hur & Liang, 2024) estimate counterfactual outcomes by pairing individuals $i, j$ with similar covariates but different treatments, and approximating the ITE as $Y_j(1) - Y_i(0)$. However, this construction implicitly assumes independence between the potential outcomes ($\rho = 0$). To our knowledge, existing matching methods do not incorporate matching mechanisms that depend directly on the value of $Y_i$.

More detailed literature review can be found in Appendix A.1.

## 3 Constructing Counterfactual Estimate under Cross-World Assumptions

Our goal is to construct a point estimate and prediction interval for the counterfactual outcome. If both $Y_i(1)$ and $Y_i(0)$ were observable for some individuals, the problem would reduce to classical regression with the factual outcome as an additional covariate. Since this is not possible, inferring counterfactual outcomes remains fundamentally challenging.

A natural starting point is to *separately* construct point estimates and prediction intervals for the treated group and the control group. For point prediction, any machine learning method, such as random forests or neural networks, can be used. For interval estimation, any conformal or other uncertainty quantification approach can be applied. We refer to Appendix A.2 for details on classical methods and their properties. Suppose their form is as follows:

$$\hat{\mu}_0(x) \text{ and } \hat{\mu}_1(x) \text{ are estimates of } \mu_0(x) \text{ and } \mu_1(x), \text{ respectively, and}$$
$$\tilde{C}_0(x) = [\hat{\mu}_0(x) - l_0(x), \ \hat{\mu}_0(x) + u_0(x)], \qquad \tilde{C}_1(x) = [\hat{\mu}_1(x) - l_1(x), \ \hat{\mu}_1(x) + u_1(x)], \tag{3}$$

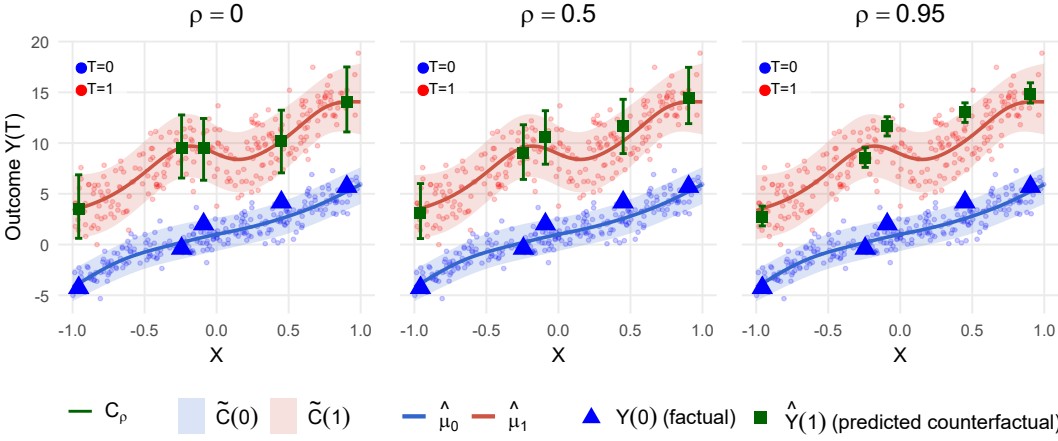

Figure 1: Proposed counterfactual estimator $\hat{Y}(1) := \hat{\mu}_\rho(x, y)$ and interval $C_\rho(x, y)$, combining baseline predictions with cross-world dependence. Here, $\rho = 0$ corresponds to ignoring the factual outcome, while $\rho = 1$ assumes perfect dependence. Illustrated on five highlighted units.

where $l_t, u_t \geq 0$ are the (lower and upper) widths of prediction intervals for $Y(t), t = 0, 1$. This is visualized in Figure 1. Ideally, $\tilde{C}_t$ satisfy either marginal or conditional coverage:

$$P\big(Y(t) \in \tilde{C}_t(X)\big) \geq 0.9, \quad \text{or} \quad P\big(Y(t) \in \tilde{C}_t(x) \mid X = x\big) \geq 0.9,$$

where marginal coverage is automatically satisfied for conformal methods, while conditional coverage typically requires large sample sizes or strong assumptions in case of high-dimensional $X$. We combine these quantities to construct a point estimate $\hat{\mu}_\rho$ and a prediction interval $C_\rho$ as follows:

**Definition 2.** *Let $\rho \in [-1, 1]$. Consider baseline estimates in the form equation 3. We first define the point predictors:*

$$\hat{\mu}_\rho^t(x, y) = \begin{cases} \hat{\mu}_1(x) + \rho \cdot \lambda(x) \cdot (y - \hat{\mu}_0(x)), & \text{if } t = 0, \\ \hat{\mu}_0(x) + \rho \cdot \dfrac{1}{\lambda(x)} \cdot (y - \hat{\mu}_1(x)), & \text{if } t = 1, \end{cases}$$

*where $\lambda(x) = \frac{l_1(x) + u_1(x)}{l_0(x) + u_0(x)}$ is the relative width of the baseline prediction intervals. Given these point predictors, we define the $C_\rho$ intervals by*

$$C_\rho^t(x, y) = \begin{cases} \Big[ \hat{\mu}_\rho^t(x, y) - \sqrt{1 - \rho^2} \cdot l_1(x), \ \hat{\mu}_\rho^t(x, y) + \sqrt{1 - \rho^2} \cdot u_1(x) \Big], & \text{if } t = 0, \\ \Big[ \hat{\mu}_\rho^t(x, y) - \sqrt{1 - \rho^2} \cdot l_0(x), \ \hat{\mu}_\rho^t(x, y) + \sqrt{1 - \rho^2} \cdot u_0(x) \Big], & \text{if } t = 1. \end{cases}$$

*For notational simplicity, we omit the superscript and write $C_\rho(x, y) = C_\rho^t(x, y)$ and $\hat{\mu}_\rho(x, y) = \hat{\mu}_\rho^t(x, y)$ when evident from context (typically when $t = 0$ and the counterfactual $Y(1)$ is of interest).*

The choices for $\hat{\mu}_\rho$ and $C_\rho$ are motivated by Theorem 1. The intuition is simple: the larger $\rho$, the more weight is put on the (centered) factual outcome. The role of $\lambda(x)$ is to adjust for potential differences in variance between treated and untreated groups; in settings where equal variances across groups can be reasonably assumed, one may simply set $\lambda(x) = 1$. While a claim of optimality in Theorem 1 is a strong statement, the result holds only under an idealized asymptotic scenario. In practice, estimation error or non-Gaussianity can lead to suboptimal performance, while additional assumptions can lead us to a different optimal prediction intervals. Nonetheless, the theorem provides valuable motivation: it shows that under ideal conditions, the $C_\rho$ construction yields the smallest valid prediction set for a counterfactual.

### 3.1 CONSISTENCY

A direct consequence of Theorem 1 is that our estimators are consistent when the cross-world dependence between $Y(1)$ and $Y(0)$ is correctly specified.

**Theorem 2** (Asymptotic consistency of $\hat{\mu}_\rho$ and $C_\rho$). *Let $x \in \mathcal{X}$ and suppose $\big(Y(1), Y(0)\big) \mid X = x$ is Gaussian with $\rho = \mathrm{cor}\big(Y(1), Y(0) \mid X = x\big) \in [-1, 1]$.*

*Let $\hat{\mu}_t(x)$ be consistent estimators of $\mu_t(x)$, and assume the prediction interval widths $l_t(x), u_t(x)$ are asymptotically conditionally valid[1] Then, for any fixed $y \in \mathbb{R}$: $\hat{\mu}_\rho(x, y)$ is a consistent estimator of the conditional mean,*

$$\hat{\mu}_\rho(x, y) \xrightarrow{p} \mathbb{E}\big[Y(1) \mid X = x, Y(0) = y\big], \quad as \ n \to \infty.$$

*The $C_\rho$ prediction intervals achieve asymptotic conditional coverage,*

$$\lim_{n \to \infty} \mathbb{P}\big(Y(1) \in C_\rho(X, Y(0)) \mid X = x, Y(0) = y\big) = 0.9.$$

The assumption of Gaussianity and $\rho(x)$ are both modeling assumptions about how $Y(1)$ and $Y(0)$ relate, and neither can be learned from data. The Gaussian copula simply translates a chosen value of $\rho(x)$ into a fully specified cross-world distribution, and any other copula could serve the same role. This highlights the central challenge of retrospective counterfactual prediction: a full dependence structure between the two potential outcomes must be specified, not estimated. Analogous consistency and optimality results to Theorem 2 can be straightforwardly derived under any alternative cross-world dependence structure.

### 3.2 Special cases: $\rho = 0$ and $\rho = 1$

When $\rho = 0$, our predictions do not depend on $y$: $\mu_\rho(x, y) = \hat{\mu}_1(x)$ and $C_\rho(x, y) = \tilde{C}_1(x)$, as the factual outcome $Y_i(0)$ provides no information about the missing potential outcome. The problem then reduces to a standard regression setting, as discussed e.g. in Lei & Candès (2021). Under $Y(1) \perp\!\!\!\perp Y(0) \mid X$, our $C_\rho$ intervals inherit the validity of the baseline $\tilde{C}_1$ interval:

$$\mathbb{P}(Y(1) \in \tilde{C}_1(X) \mid X = x) \geq 0.9 \implies \mathbb{P}(Y(1) \in C_\rho(X, Y(0)) \mid X = x, Y(0) = y) \geq 0.9. \quad (4)$$

Moreover, $C_\rho$ is marginally valid even in finite samples, if $\tilde{C}_1$ is marginally valid (which holds if a conformal method is used).

When $\rho = \pm 1$ and $\lambda(x) = 1$, we have $\hat{\mu}_\rho(x, y) = y + \hat{\tau}(x)$ and $C_\rho(x, y) = \{\hat{\mu}_\rho(x, y)\}$, corresponding to a constant treatment effect:

$$\mu_\rho(x, y_0) = \hat{\mu}_\rho(x, y_0) \implies \mathbb{P}(Y(1) \in C_\rho(X, Y(0)) \mid X = x, Y(0) = y) = 1. \quad (5)$$

In practice, however, $\mu_\rho(x, y_0)$ is unknown and must be estimated, introducing bias and potentially non-valid prediction intervals. Section 3.3 discusses how to extend $C_\rho$ intervals to account for the additional uncertainty from this estimation.

### 3.3 Finite sample bias correction: introducing $C_\rho^{+CI}$ prediction intervals

We enlarge $C_\rho$ *prediction intervals* by adding *confidence intervals* for $\mu_\rho$, estimated for instance via bootstrapping.

**Definition 3.** *Let $\rho \in [-1, 1]$. Consider prediction intervals for $Y(1)$ and $Y(0)$ of the form equation 3, and suppose we have confidence intervals $CI(x, y) = [\hat{\mu}_\rho(x, y) - r_l(x, y), \hat{\mu}_\rho(x, y) + r_u(x, y)]$. We define the bias-corrected $C_\rho^{+CI}$ intervals as*

$$C_\rho^{+CI}(x, y) = \Big[ \hat{\mu}_\rho(x, y) - c \cdot r_l(x, y) - \sqrt{1 - \rho^2} \cdot l_{1-T_i}(x), \ \ \hat{\mu}_\rho(x, y) + c \cdot r_u(x, y) + \sqrt{1 - \rho^2} \cdot u_{1-T_i}(x) \Big],$$

*where $l_{1-T_i}(x)$ and $u_{1-T_i}(x)$ select the appropriate prediction bounds depending on treatment status $T_i$, and $c \in [0, 1]$ is a hyperparameter. In simple terms, $C_\rho^{+CI}$ extends $C_\rho$ by adding a scaled confidence interval around $\hat{\mu}_\rho(x, y)$, with scaling factor $c$. We consider the choice $c = \rho^2$ following the same argument as in (Bodik et al., 2025).*

---

[1]This holds for many nonparametric estimators under mild smoothness assumptions, including random forests for estimating $\hat{\mu}_t(x)$ and CQR using quantile random forests for prediction intervals. More details are given in Appendix D.

Following equation 4 and equation 5, when $\rho = 0$, no adjustment is needed, while for $\rho = \pm 1$, full confidence intervals must be incorporated to guarantee correct coverage. This motivates the choice $c = \rho^2$, ensuring that $C_\rho^{+CI}$ smoothly interpolates between no correction ($\rho = 0$) and full correction ($\rho = 1$). For this choice, we also have the following guarantee.

**Consequence 1.** *If $\rho = \pm 1$ and confidence intervals satisfy $\mathbb{P}(\mu(x, y_0) \in \hat{\mu}_\rho(x, y_0) \pm r(x, y_0)) \geq 1 - \alpha$, then $\mathbb{P}(Y(1) \in C_\rho^{+CI}(X, Y(0)) \mid X = x, Y(0) = y) \geq 1 - \alpha$.*

## 4 Numerical experiments

We evaluate our method on synthetic, semi-synthetic, and real datasets using both point estimation and prediction interval metrics, comparing against four baselines under varying cross-world correlation $\rho$. A user-friendly implementation of our methods in both R and Python, along with scripts to reproduce all experiments, is available at: [*github link anonymized for review*].

### 4.1 Details

**Datasets:** We consider a variety of data-generating processes commonly used in the related literature; full details are provided in Appendix C.1. The **synthetic** datasets feature non-constant propensity scores and randomly generated CATE functions based on smooth random polynomials. These settings allow us to vary the dimensionality $d = \dim(\mathbf{X})$ and the cross-world correlation parameter $\rho$, thus controlling both complexity and treatment-effect heterogeneity. In addition, we include the **IHDP** dataset, which uses real covariates from a randomized trial and simulated counterfactual outcomes, providing a semi-synthetic benchmark. The **Twins** dataset contains real covariates and real paired outcomes corresponding to different treatment assignments, enabling the construction of both factual and counterfactual outcomes for each unit.

**Implementation details:** To better reflect real-world scenarios where $\rho$ is unknown, we report both i) $\rho_{used} = \rho_{true}$ and ii) $\rho_{used} = \rho_{true} + Unif(-0.5, 0.5)$ capped at $[-1, 1]$.

To construct the proposed $C_\rho$ and $C_\rho^{+CI}$ intervals, we use CQR (see Appendix A.2) to produce the base intervals in equation 3. While more advanced methods often achieve better empirical results, we adopt CQR as a simple, well-established baseline, following Lei & Candès (2021); Alaa et al. (2023), and Bodik et al. (2025).

Our algorithm jointly estimates conditional means and quantiles: in low dimensions ($d \leq 5$) we use GAM for the mean and qGAM (Fasiolo et al., 2017) for quantiles, while in higher dimensions ($d > 5$) we switch to random forests for the mean and quantile random forests (Meinshausen & Ridgeway, 2006) for quantiles, trading some low-dimensional efficiency for scalability. TabPFN (Hollmann et al., 2023) is a good potential alternative.

**Baseline methods:** In Appendix A.1, we provide details of the existing methods used to estimate counterfactuals. We consider four representative approaches. First, **CATE-adjusted imputation** estimates the CATE via a T-learner (Künzel et al., 2019), DR-learner (doubly robust, Dukes et al. (2024)) or Generalized Random Forest (Athey et al., 2019), and adjusts the observed outcome using $\hat{Y}_i(1) = Y_i(0) + \hat{\tau}(X_i)$. We only report the T-learner as it yielded the best results on the considered datasets. Note that while many other CATE estimators exist, the goal is to illustrate the core imputation approach, which remains fundamentally limited even with perfectly estimated CATE. Second, **Direct Outcome (DO) modeling** fits the treatment-specific regression $\hat{Y}_i(1) := \mu_1(X_i)$ using Random Forests (Wager & Athey, 2018) or Generalized Additive Models (Fasiolo et al., 2017) (using the same choices as in $C_\rho$). Third, **Matching-based imputation** uses nearest-neighbor matching with Mahalanobis distance to impute the missing potential outcome from similar units in the opposite treatment group. Fourth, **adversarial generative modeling** employs GANITE (Yoon et al., 2018), a two-stage generative adversarial network that imputes and refines counterfactual predictions, typically suitable only in high-dimensional, nonlinear settings.

**Setup**: We conducted experiments on datasets: synthetic ($n = 1000$), IHDP ($n = 747$), and Twins ($n = 11{,}983$). Each synthetic and IHDP experiment was repeated 50 times to reduce Monte Carlo variability, while the Twins dataset was analyzed once using the full sample. All methods used an 80/20 train–calibration split for CQR and prediction intervals at level $\alpha = 0.1$. Computing $\mu_\rho$ and $C_\rho$ is fast, as the main cost lies in fitting four quantile regressions; however, $C_\rho^{+CI}$ requires

378
379
380
381
382
383
384
385
386
387
388
389
390
391
392
393
394
395
396
397
398
399
400
401

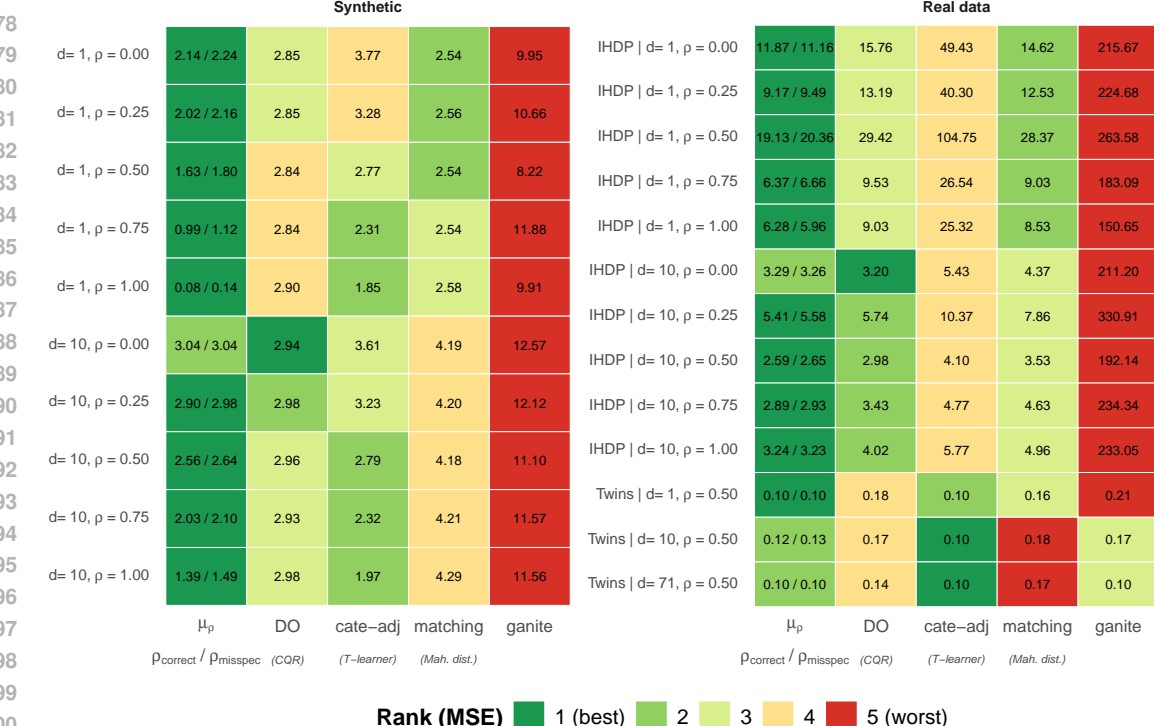

**Figure 2:** Mean squared error of different estimators across different datasets, averaged over 50 repetitions. In $\mu_\rho$, we use either $\rho = \rho_{true}$, or mimic misspecification by using $\rho = \rho_{true} + Unif(-0.5, 0.5)$. Standard deviations for each entry can be found in Appendix B.

computing bootstrap confidence intervals (we used 100 bootstraps), which is computationally more intensive; running all datasets and repetitions took approximately four days on an Intel Core i5-6300U (2.5 GHz, 16 GB RAM).

**Metrics:** To assess performance, we use MSE for point predictions and the Interval Score (metric that combines coverage and width) for prediction intervals:

$$\text{MSE} = \frac{1}{n} \sum_{i=1}^{n} (\hat{Y}_i^{cf} - Y_i^{cf})^2, \qquad \text{IS}_\alpha = \frac{1}{n} \sum_{i=1}^{n} (U_i - L_i) + \frac{2}{\alpha} \left[ (L_i - Y_i^{cf})_+ + (Y_i^{cf} - U_i)_+ \right],$$

where $[L_i, U_i]$ are the estimated prediction intervals at level $1 - \alpha$ and $z_+ = \max(z, 0)$.

### 4.2 RESULTS OF THE EXPERIMENTS

Figure 2 presents the MSE results of point predictions; Figure 5 in Appendix C.2 presents the interval scores for prediction intervals. Both of the variants (correctly specified $\rho$ and misspecified $\rho$) strongly outperform other methods in scenarios where $\rho \neq 0$ or 1; if $\rho = 0$ note that DO have almost identical performance as our method. If $\rho = 1$, the CATE-adjusted estimators have competitive performance.

While it seems that GANITE has very bad performance, note that it was built for large dimensional problems, and for large $d$ and $n$ it would perform often better. Our method is more suitable for low dimensions, when the factual $Y(T)$ contains significant information beyond the information in the observed covariates.

In a few real-world datasets, $\rho_{\text{misspec}}$ yields slightly better performance than $\rho_{\text{correct}}$, a consequence of Monte Carlo variability. As shown in Figure 5, the corresponding confidence intervals are large in these cases, and resolving these differences would require hundreds of repetitions to reduce simulation noise.

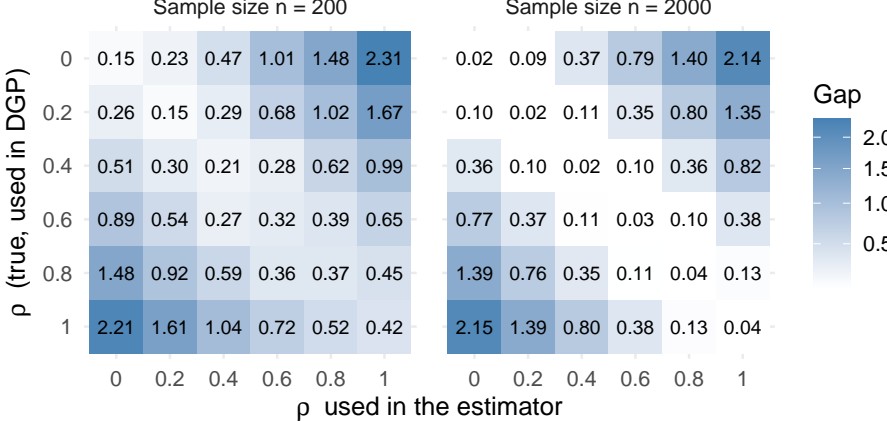

Figure 3: Gap $= \mathrm{MSE}_{\mathrm{our}} - \mathrm{MSE}_{\mathrm{oracle}}$ calculated across different misspecifications of $\rho$. Bias persists when $\rho$ is far from the truth, but vanishes asymptotically if $\rho$ is specified correctly. This demonstrates that incorporating even approximate knowledge of cross-world dependence improves counterfactual predictions.

### 4.3 ADDITIONAL EXPERIMENTS: MISSPECIFIED $\rho$ AND NON-GAUSSIANITY

We conduct two additional experiments, evaluating the Gap $= \mathrm{MSE}_{\mathrm{our}} - \mathrm{MSE}_{\mathrm{oracle}}$, where the oracle estimator is equal to the true $\mathbb{E}[Y^{cf} \mid X, Y^{obs}, T]$. All details can be found in Appendix B.

- **(Misspecifying $\rho$).** Figure 3 reports experiments on synthetic data varying the *true* correlation $\rho_{\mathrm{true}}$ in the data-generating process (DGP) and the *assumed* value $\rho_{\mathrm{est}}$ in our estimator. Bias grows with misspecification $|\rho_{\mathrm{est}} - \rho_{\mathrm{true}}|$, and vanishes with larger $n$ only when $\rho_{\mathrm{est}}$ is close to $\rho_{\mathrm{true}}$; otherwise, it persists even asymptotically. This shows that even rough knowledge of $\rho$ yields large gains over ignoring the factual outcome.

- **(Robustness to non-Gaussianity).** Appendix B.1 (Figure 4) contains experiments with non-Gaussian outcome distributions $(Y(0), Y(1))$. In all cases the gap vanishes with $n$, though convergence is slower under non-Gaussian noise. Discrepancies are most visible at $\rho = 1$.

## 5 CONCLUSION AND FUTURE RESEARCH

The factual outcome carries valuable individual-level information that should not be ignored in counterfactual prediction. We formalize the importance of the factual outcome through the cross-world correlation parameter $\rho$, which determines how strongly observed and unobserved outcomes are linked. By treating $\rho$ as an explicit modeling choice, our approach interpolates between classical extremes, with $\rho = 0$ discarding the factual outcome and $\rho = 1$ assuming constant effects, and delivers predictions that are theoretically well motivated and empirically effective whenever even approximate knowledge of $\rho$ is available.

Although $\rho$ is not identifiable from observed data, *every existing method already makes a fixed, implicit assumption about $\rho$.* Our contribution is to make this dependence explicit, enabling practitioners to incorporate domain knowledge or sensitivity analysis into counterfactual inference. This transparency clarifies the assumptions underlying prediction and opens new possibilities for modeling cross-world dependence.

Future work should explore richer dependence structures, such as copula-based models, which would enable a broader class of assumptions about how potential outcomes co-vary. This would yield a more general framework for counterfactual prediction, accommodating settings where simple correlation is inadequate. Another promising direction is to extend the methodology to continuous treatments or dynamic settings such as time series, where cross-world assumptions could provide structure for dose–response curves or evolving interventions, thereby enhancing both interpretability and stability. Beyond methodological extensions, future research may also investigate applications in domains where expert knowledge about cross-world dependence is available, such as medicine, economics, or climate science.

## Reproducibility Statement and Usage of Large Language Models

All code and datasets used in this work are provided in the supplementary material to ensure full reproducibility of our results. We declare that we used a large language model for grammar and language polishing, as well as for limited coding assistance (e.g., boilerplate code and debugging). All conceptual and theoretical contributions, experimental designs, and conclusions are our own.

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

# Appendix

## A  LITERATURE REVIEW: DETAILS

### A.1  COUNTERFACTUAL ESTIMATION METHODS: OTHER APPROACHES

We consider four classes of approaches for estimating the unobserved potential outcome $Y_i(1)$ for units with $T_i = 0$ (and analogously $Y_i(0)$ for $T_i = 1$).

**CATE-adjusted imputation (*CATE-adj*).**  This approach first estimates CATE $\tau(X_i)$ and then shifts the observed control outcome by this estimated effect:

$$\hat{Y}_i(1) = Y_i(0) + \hat{\tau}(X_i).$$

We use three alternative CATE estimators: the T-learner (Künzel et al., 2019), the Generalized Random Forest (GRF) (Athey et al., 2019), and a doubly robust (DR) estimator (Dukes et al., 2024). Closely related meta-learners include the S-learner, which fits a single model with treatment as an input feature, and the X-learner, which augments the T-learner with imputed treatment effects for the opposite treatment group and often performs well under treatment imbalance (Künzel et al., 2019). These alternative meta-learners share the same conceptual foundation. Johansson et al. (2016); Lacombe & Sebag (2025) use deep learning alternatives; balancing counterfactual regression or adding assymetrical latend represnetation.

To quantify uncertainty, confidence intervals are computed using standard procedures, obtaining prediction intervals in a form $\hat{Y}_i(1) = Y_i(0) + \hat{\tau}(X_i) \pm conf.int(\hat{\tau}(X_i))$. In our experiments, we only considered T-learner, GRF and DR estimators for CATE-adjusted imputation, as other approaches are typically significantly more performative only in high-dimensional datasets or when treated and untreated units differ substantially, which is not the case in our datasets.

**Direct outcome modeling (*DO*).**  Here we model the treatment-specific regression function $\mu_1(x) = \mathbb{E}[Y \mid X = x, T = 1]$ directly from the treated sample and use $\hat{Y}_i(1) = \hat{\mu}_1(X_i)$ for counterfactual prediction. We consider two implementations: Random Forests (RF) (Wager & Athey, 2018) and Generalized Additive Models (GAM) (Fasiolo et al., 2017). Unlike the CATE-adjusted approach, these methods do not require access to the observed control outcome $Y_i(0)$ for the unit, relying entirely on model-based extrapolation from treated units. To quantify uncertainty, we use the same prediction intervals as in equation 3.

There is also a large number of similar approaches besides RF and GAM, also adjusting for the distribution shift between the treated/untreated groups. Yao et al. (2018) employ deep representation learning to estimate $\hat{Y}_i(1-T) = g(f(X_i), T_i)$ where $f, g$ are neural networks based preserving local similarity between the treated groups.

**Matching-based imputation (*Matching*).**  This approach imputes missing potential outcomes using outcomes from similar units in the opposite treatment group, selected via a distance metric in covariate space (Stuart, 2010; Abadie & Imbens, 2006). Beyond nearest-neighbor and optimal matching, advances include kernel-based matching to minimize estimation error (Kallus, 2017) and full or genetic matching combined with double-robust analysis for improved bias and efficiency (Colson et al., 2016). For high-dimensional or categorical data, algorithms like DAME prioritize relevant covariates (Dieng et al., 2019). Similar ideology was also used in ALRITE (Lacombe & Sebag, 2025), where the authors imputed counterfactuals based on the closest distance in a latent space, in order to improve CATE estimation.

We implemented nearest-neighbor matching with a uniform kernel and optional replacement, using either the Mahalanobis distance between standardized covariates or the absolute difference in logit propensity scores (the former led to better results so we only report that). The propensity scores is estimated by standard classification forest. For a treated unit, the counterfactual $\hat{Y}_i(0)$ is the average outcome among its matched controls, and vice versa for control units. This nonparametric approach relies on local overlap in covariates and assumes conditional independence of potential outcomes and treatment given covariates. To quantify uncertainty, we construct unit-level prediction

intervals for the counterfactuals using the empirical variance of the donor outcomes: for a unit with $K \geq 2$ matches, the half-width is given by $t1 - \alpha/2, K - 1 \cdot s/\sqrt{K}$, where $s$ is the sample standard deviation of the matched donor outcomes, yielding $(\hat{Y}_i^{\mathrm{cf}} \pm$ half-width); if $K = 1$, the half-width is zero. This approach implicitly assumes conditional independence of potential outcomes ($\rho = 0$, similarly to DO) and independent treatment given covariates.

**Adversarial generative modeling (*GANITE*).** GANITE (Yoon et al., 2018) employs a two-stage generative adversarial network (GAN) framework tailored to causal inference. In the first stage, a generator–discriminator pair is trained to impute the missing counterfactual outcomes by making the generated outcomes indistinguishable from observed ones given covariates and treatment assignment. In the second stage, a separate adversarial network refines these predictions to improve estimation of individualized treatment effects, encouraging accurate recovery of both potential outcomes simultaneously. This approach is particularly suited to high-dimensional, nonlinear settings. Some extentions were also proposed that work better under some alternative scenarios (e.g. SCIGAN-ITE by Bica et al. (2020)).

**Other approaches.** Some other approaches exist, such as **Bayesian causal inference**, where the missing counterfactuals are treated as latent variables, and uncertainty is integrated through the posterior distribution. For example, Alaa & van der Schaar (2017) propose a Bayesian multitask Gaussian process to jointly model $\big(Y(1), Y(0)\big) \mid X$, producing posterior distributions over the potential outcomes. While Bayesian methods offer coherent uncertainty quantification, they often rely on strong modeling assumptions and can be sensitive to prior specifications (Li et al., 2022). Moreover, they can be restrictive when aiming to leverage flexible modern machine learning techniques.

### A.2 UNCERTAINTY QUANTIFICATION AND PREDICTION INTERVALS IN CLASSICAL REGRESSION

In a standard regression framework, we observe data $(X_i, Y_i) \sim P_X \times P_{Y|X}$ for $i = 1, \ldots, n$, and seek a prediction set $C(X)$ for future responses that satisfies a coverage property. Two common notions of coverage are:

$$\mathbb{P}\big(Y_{n+1} \in C(X_{n+1})\big) \geq 1 - \alpha \qquad \text{(marginal coverage)},$$
$$\mathbb{P}\big(Y_{n+1} \in C(X_{n+1}) \mid X_{n+1} = x\big) \geq 1 - \alpha \qquad \text{(conditional coverage)}.$$

Conditional coverage is a stronger requirement but is generally unattainable in a distribution-free, finite-sample setting without strong assumptions or asymptotics (Barber et al., 2020). By contrast, marginal coverage can be attained without modeling assumptions via conformal prediction (Angelopoulos et al., 2024). Recent work has also explored data-driven techniques to improve conditional coverage, such as combining epistemic+aleatoric sources of uncertainty (Azizi et al., 2025), rectifying conformity scores (Plassier et al., 2025), or optimizing subgroup-conditional guarantees through flexible frameworks like Kandinsky conformal prediction (Bairaktari et al., 2025). These developments are consistent with the broader principles of Predictability, Computability, and Stability (PCS) advocated for trustworthy data science (Agarwal et al., 2025; Yu & Barter, 2024).

Conformal methods produce prediction intervals with exact finite-sample marginal coverage under exchangeability of the observed and future data points (Vovk et al., 2005; Angelopoulos et al., 2024). These methods typically split the data into training and calibration subsets, construct a preliminary predictor on the training set, and adjust it on the calibration set to guarantee coverage. A prominent example is Conformalized Quantile Regression (CQR), which uses estimated conditional quantiles to build tighter prediction intervals (Romano et al., 2019).

**Estimation procedure for CQR.** The key idea of CQR is to combine quantile regression with conformal calibration:

1. **Split the data.** Randomly divide the dataset into a training set $\mathcal{D}_{\mathrm{train}}$ and a calibration set $\mathcal{D}_{\mathrm{calib}}$. The split fraction is typically 80/20.

2. **Fit quantile regression models.** On $\mathcal{D}_{\mathrm{train}}$, estimate the conditional lower and upper quantile functions $\hat{q}_{\alpha/2}(x)$ and $\hat{q}_{1-\alpha/2}(x)$, often quantile random forest (Meinshausen & Ridgeway, 2006), qGAM (Fasiolo et al., 2017) or neural networks to approximate conditional quantiles for levels $\alpha/2$ and $1 - \alpha/2$.

3. **Compute conformity scores.** For each $(X_i, Y_i) \in \mathcal{D}_{\text{calib}}$, compute the nonconformity score:

$$s_i = \max\{\hat{q}_{\alpha/2}(X_i) - Y_i, \; Y_i - \hat{q}_{1-\alpha/2}(X_i), \; 0\}.$$

This measures how far $Y_i$ lies outside the estimated conditional quantile interval.

4. **Calibrate using empirical quantiles.** Let $Q_{1-\alpha}(s_1, \ldots, s_m)$ be the $(1 - \alpha)$-empirical quantile of the scores from the calibration set ($m = |\mathcal{D}_{\text{calib}}|$).

5. **Construct prediction intervals.** For a new point $x$, the CQR prediction set is:

$$\tilde{C}(x) = \left[\hat{q}_{\alpha/2}(x) - Q_{1-\alpha}, \; \hat{q}_{1-\alpha/2}(x) + Q_{1-\alpha}\right].$$

This adjustment ensures that the final interval achieves marginal coverage at level $1 - \alpha$ in finite samples under exchangeability, while leveraging conditional quantile estimates for tighter intervals.

However, exchangeability (slightly weaker assumption than i.i.d.) can fail in the presence of covariate shift, e.g., in observational studies comparing treated and untreated units. In such settings, even defining marginal coverage requires specifying the *target covariate distribution*: should coverage be with respect to $P_{X|T=1}$ (treated), $P_{X|T=0}$ (untreated), or a mixture $P_X$? This point is emphasized in Lei & Candès (2021). If one could attain conditional coverage, covariate shift would not pose a problem (recall that conditional coverage implies marginal coverage under any $P_X$) but such guarantees remain scarce (Gibbs et al., 2025).

To address distributional shift, weighted conformal prediction adjusts calibration via importance weights derived from the likelihood ratio between covariate distributions; when this ratio is known, one can guarantee exact marginal coverage for the chosen target population (Tibshirani et al., 2019). When the ratio (or propensity score $\pi(x)$) is estimated, asymptotically valid marginal coverage is still achievable, with strong empirical performance (Lei & Candès, 2021). Recent approaches refine this idea by incorporating likelihood-ratio regularization for high-dimensional covariates (Joshi et al., 2025) or leveraging unlabeled test data to adapt coverage under label scarcity (Kasa et al., 2025). For settings with both covariate shift and posterior drift, weighted conformal classifiers have been proposed (Wang & Qiao, 2025).

# B ADDITIONAL EXPERIMENTS: MISSPECIFIED $\rho$ AND NON-GAUSSIANITY

## B.1 HOW VITAL IS THE ASSUMPTION OF GAUSSIANITY?

We evaluate the sensitivity of our counterfactual estimation method to violations of the Gaussianity assumption in the joint distribution of potential outcomes. Specifically, we use the Synthetic dataset described in Appendix C.2, but replace the Gaussian error terms with non-Gaussian marginals coupled through different copulas. Formally, for each unit $i$, we generate

$$(\varepsilon_i^0, \varepsilon_i^1) \overset{i.i.d}{\sim} \mathrm{Copula}_\rho(F_0, F_1),$$

where $F_t$ denotes the marginal distribution of $\varepsilon_i^t$ (e.g., $t = 0, 1$ could follow Student-$t$, Laplace, or Chi-square distributions), and $\mathrm{Copula}_\rho$ is a copula with correlation $\rho$. By Sklar's theorem, this ensures that the joint distribution of $(\varepsilon_i^0, \varepsilon_i^1)$ has the specified marginals while preserving the desired correlation structure through $\mathrm{Copula}_\rho$. We experiment with Gaussian and Gumbel copulas to capture symmetric as well as asymmetric dependence patterns.

We vary the following factors:

- Marginal distributions: Gaussian, Student-$t$ ($df = 3$), Laplace, and Chi-square ($df = 3$),
- Copula families: Gaussian and Gumbel,
- Cross-world correlation: $\rho \in \{0, 0.5, 1\}$,
- Sample size: $n \in \{100, 300, 500, 2000\}$ with covariate dimension fixed at $d = 1$.

For each configuration, we generate 50 replications and compare our estimate $\hat{\mu}_\rho$ against the oracle estimator

$$\hat{Y}_{\mathrm{oracle}}^{cf} := \mathbb{E}[Y^{\mathrm{cf}} \mid X, Y^{\mathrm{obs}}, T],$$

which leverages the true joint distribution. We report the performance gap

$$\mathrm{Gap} = \mathrm{MSE}_{\mathrm{our}} - \mathrm{MSE}_{\mathrm{oracle}}, \qquad \mathrm{MSE}_{\mathrm{our}} = \frac{1}{n}\sum_{i=1}^{n}(\hat{Y}_i^{cf} - Y_i^{cf})^2, \qquad \hat{Y}_i^{cf} = \hat{\mu}_\rho.$$

Figure 4 summarizes the results. In all cases, the gap decreases with $n$, demonstrating that our estimator converges to the oracle regardless of the marginal distribution or copula. The effect of non-Gaussianity is therefore limited to finite samples: convergence is noticeably slower under heavy-tailed or skewed marginals, particularly when $\rho = 1$, but the asymptotic behavior remains unchanged. By contrast, under independence ($\rho = 0$), our estimator is nearly indistinguishable from the oracle even in small samples.

**In conclusion, violations of Gaussianity do not seem to threaten the validity of our method, but they can slow finite-sample convergence; especially under large cross-world dependence.**

## B.2 DETAILS ABOUT FIGURE 3 AND MISSPECIFIED $\rho$

To study the effect of misspecifying the cross-world correlation $\rho$, we carried out a grid experiment on synthetic data. For each design point, we distinguish between the **true** value $\rho_{\mathrm{true}}$ used in the data-generating process (DGP), and the **assumed** value $\rho_{\mathrm{est}}$ used in our estimator $\hat{\mu}_\rho$.

We consider the *synthetic* dataset (see Section C.1), a univariate covariate setting ($d = 1$), two sample sizes ($n = 200$ and $n = 2000$), and repeated each experiment 50 times to reduce Monte Carlo variability. The true correlation $\rho_{\mathrm{dgp}}$ was varied over a grid $\{0, 0.1, \ldots, 1\}$, and for each value we estimated counterfactuals under a grid of assumed correlations $\rho_{\mathrm{est}} \in \{0, 0.1, \ldots, 1\}$.

For each pair $(\rho_{\mathrm{true}}, \rho_{\mathrm{est}})$, we generated synthetic data, computed counterfactual estimates with our method using $\rho_{\mathrm{est}}$, and compared performance against the oracle estimator $\mathbb{E}[Y^{cf} \mid X, Y^{obs}, T]$. We measured performance using the mean squared error (MSE) of counterfactual predictions, and summarized results via the $\mathrm{Gap} = \mathrm{MSE}_{\mathrm{our}} - \mathrm{MSE}_{\mathrm{oracle}}$. Results (Figure 3) show that the gap increases systematically with the degree of misspecification $|\rho_{\mathrm{est}} - \rho_{\mathrm{true}}|$. When the assumed correlation is close to the truth, the gap shrinks as $n$ grows, and bias vanishes asymptotically. In contrast, for

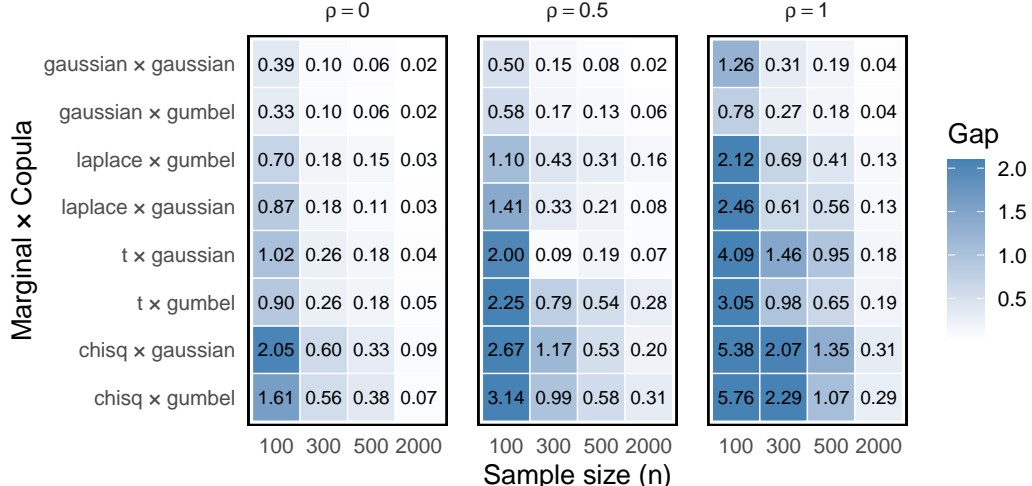

Figure 4: Gap $= \mathrm{MSE}_{\mathrm{our}} - \mathrm{MSE}_{\mathrm{oracle}}$ calculated across different marginal–copula distributions of potential outcomes $\big(Y(0), Y(1)\big)$. Here, we only considered correctly specified $\rho$ in the estimation.

larger misspecifications, the bias persists even at large $n$, indicating that asymptotic consistency requires $\rho_{\mathrm{est}} \approx \rho_{\mathrm{true}}$. These results show the importance of approximate domain knowledge of $\rho$: even approximate information about its value can yield large gains over methods that implicitly assume $\rho = 0$ or $\rho = 1$.

## C  APPENDIX: NUMERICAL EXPERIMENTS

We provide full details about our experiments below.

### C.1  DATASETS

We investigate three types of data-generating mechanisms:

- **Synthetic** (taken from (Bodik et al., 2025)): For the univariate case ($d = 1$), we draw $X \sim \text{Unif}(-1, 1)$. When $d > 1$, we follow the setup in Wager & Athey (2018); Alaa et al. (2023); Lei & Candès (2021); Jonkers et al. (2024) and generate covariates $\mathbf{X} = (X_1, \ldots, X_d)$, where each $X_j = \Phi(\tilde{X}_j)$ and $\Phi$ is the standard normal CDF. The latent vector $(\tilde{X}_1, \ldots, \tilde{X}_d)$ is sampled from a multivariate Gaussian distribution with zero mean and constant pairwise correlation $\text{Cov}(\tilde{X}_j, \tilde{X}_{j'}) = 0.25$ for $j \neq j'$. Treatment assignments are drawn from a propensity score function

$$\pi(\mathbf{X}) = \frac{1 + |X_1|}{4} \in [0.25, 0.5],$$

ensuring adequate overlap. The potential outcomes are defined as

$$Y_i(0) = f_0(\mathbf{X}_i) + \varepsilon_i^0,$$
$$Y_i(1) = f_0(\mathbf{X}_i) + \tau(\mathbf{X}_i) + \varepsilon_i^1,$$

with noise terms jointly distributed as

$$(\varepsilon_i^0, \varepsilon_i^1) \sim \mathcal{N}\left( \begin{bmatrix} 0 \\ 0 \end{bmatrix}, \begin{bmatrix} 1 & 2\rho \\ 2\rho & 4 \end{bmatrix} \right).$$

  The treatment effect function $\tau(\mathbf{x}) = \tau(x_1, x_2)$ is a smooth random polynomial depending on the first two covariates (or only on $x_1$ when $d = 1$), generated using a Perlin noise generator (Perlin, 1985) following Bodik & Chavez-Demoulin (2025). The baseline function is $f_0(x) = \beta^\top x$ with $\beta$ drawn from a standard normal distribution.

- **IHDP (semi-synthetic):** Originally introduced in Hill (2011), this dataset contains 25 pre-treatment covariates (e.g., birth weight, maternal age, education level) denoted by $\mathbf{X}$. The binary treatment $T$ indicates whether the infant participated in the intervention program. Potential outcomes represent cognitive test scores, were simulated in Hill (2011) as

$$Y_i(0) = f_0(X_i) + \varepsilon_i^0, \tag{6}$$
$$Y_i(1) = f_1(X_i) + \varepsilon_i^1, \tag{7}$$

  where $\varepsilon_i^0, \varepsilon_i^1 \overset{\text{i.i.d.}}{\sim} N(0, 1)$. The functions $f_0$ and $f_1$ are either random linear (case "A") or nonlinear (case "B"). We only consider case "B".

  While the original setup fixes $\rho = 0$, we also consider a correlated noise version:

$$\begin{pmatrix} \varepsilon_i^0 \\ \varepsilon_i^1 \end{pmatrix} \overset{\text{i.i.d.}}{\sim} \mathcal{N}\left( \begin{pmatrix} 0 \\ 0 \end{pmatrix}, \begin{pmatrix} 1 & \rho \\ \rho & 1 \end{pmatrix} \right).$$

  which better reflects empirical situations in which the two potential outcomes are not independent but share substantial underlying information.

- **Twins (real-world):** We use the U.S. twin birth records (1989–1991) described in Louizos et al. (2017), restricted to same-sex twins with both birth weights below $2\,\text{kg}$. Each pair comes with detailed perinatal covariates, including maternal risk factors, prenatal care indicators, and demographic information. In this context, twins are viewed as natural counterfactuals for one another, so the potential outcomes can be conceptually "observed" by comparing mortality for the heavier twin ($T = 1$) and the lighter twin ($T = 0$) within the same pair. The outcome variable is one-year mortality. In our analysis, we work with a balanced sample containing a moderate number of individuals and a small set of covariates, obtained after standard preprocessing.

## C.2 Interval Scores Results: Use $C_\rho$ for $\rho \le 0.5$ and $C_\rho^{+CI}$ for $\rho > 0.5$

Figures 5 and 6 report the Interval Scores (IS) of the competing methods across all datasets considered in our experiments. The Interval Score jointly evaluates interval width and coverage, with lower values indicating more efficient and reliable prediction intervals. While GANITE is excluded from these comparisons because it does not provide prediction intervals out of the box, one could imagine extending it with Bayesian or conformalized post-processing layers to quantify uncertainty. For instance, sampling-based approaches could be added to its adversarial generator, or conformal calibration could be applied on top of GANITE outputs. However, such adaptations are not standard, and we therefore omit GANITE from the interval score plots.

**Results.** When using the bias-corrected $C_\rho^{+CI}$ variant, our method achieves consistently strong results, typically outperforming all baselines across datasets. The only exception is when $\rho = 0$, in which case Direct Outcome (DO) estimators attain nearly identical performance. The main drawback of $C_\rho^{+CI}$ lies in its computational cost, since constructing bootstrap confidence intervals is substantially more demanding than computing $C_\rho$. Moreover, when $\rho$ is large, estimation error in $\hat{\mu}_\rho$ can induce bias, leading to undercoverage and consequently poor Interval Scores. In practice, we therefore recommend using the uncorrected $C_\rho$ intervals when $\rho \le 0.5$, while for $\rho > 0.5$ the bias-corrected $C_\rho^{+CI}$ intervals are preferable, as they yield the greatest empirical gains.
**Recommendation: $C_\rho$ is satisfactory if $\rho \le 0.5$, and ideally use $C_\rho^{+CI}$ if $\rho > 0.5$.**

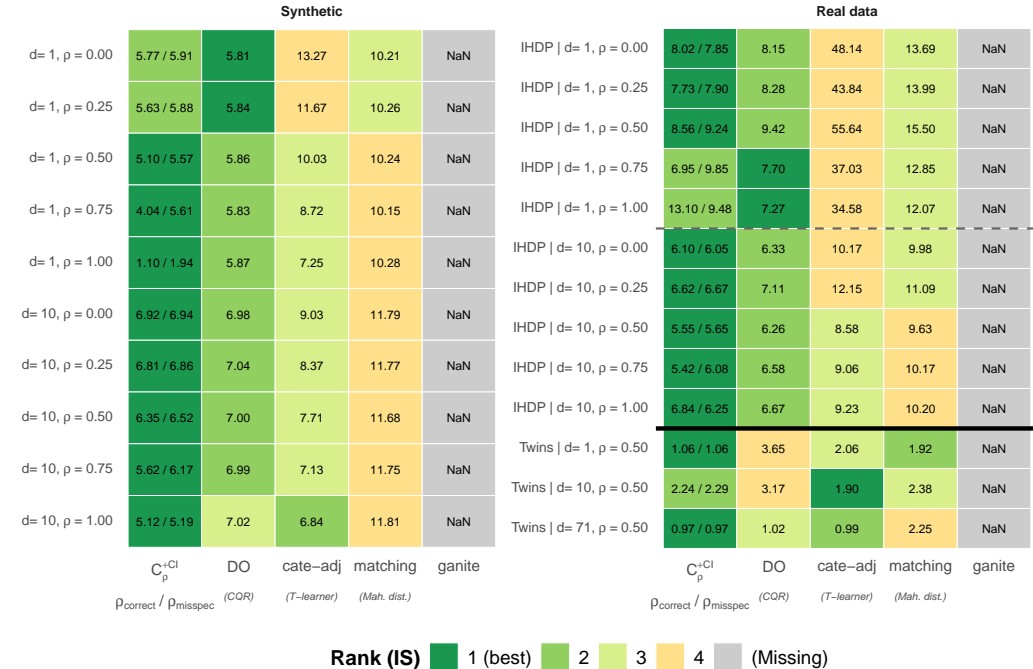

Figure 5: Interval Scores of different prediction interval methods across all datasets. Here, $C_\rho^{+CI}$, the bias-corrected version of $C_\rho$ introduced in Section 3.3, is used. GANITE is excluded since it does not provide a natural way of constructing prediction intervals.

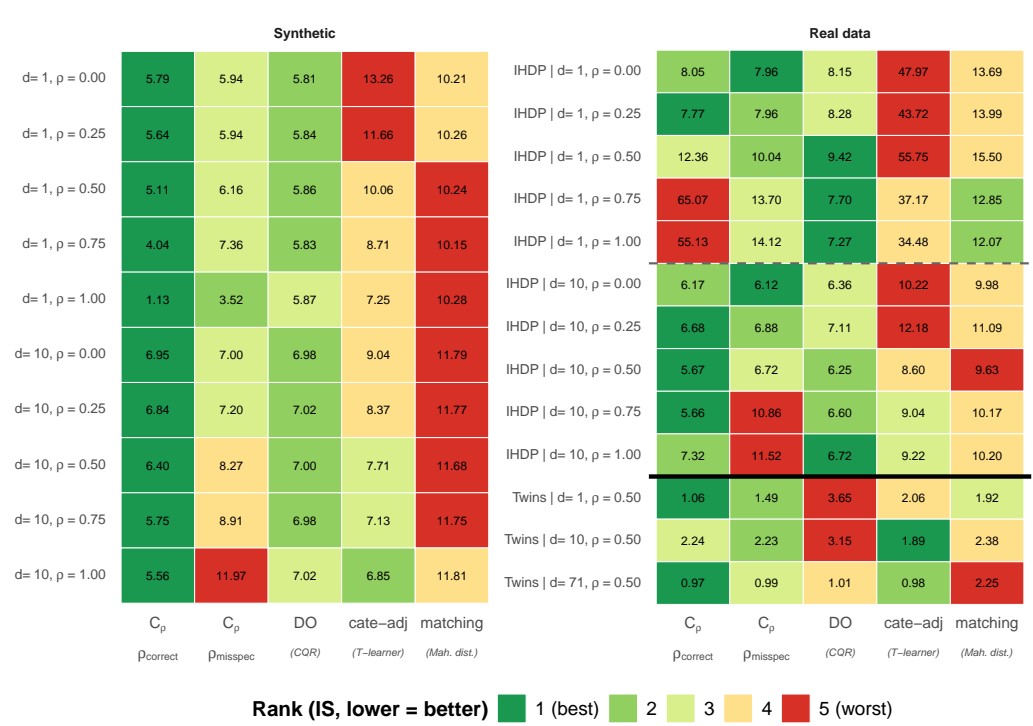

Figure 6: Interval Scores of different prediction interval methods across all datasets. Here, the uncorrected $C_\rho$ intervals, as defined in Section 3, are used.

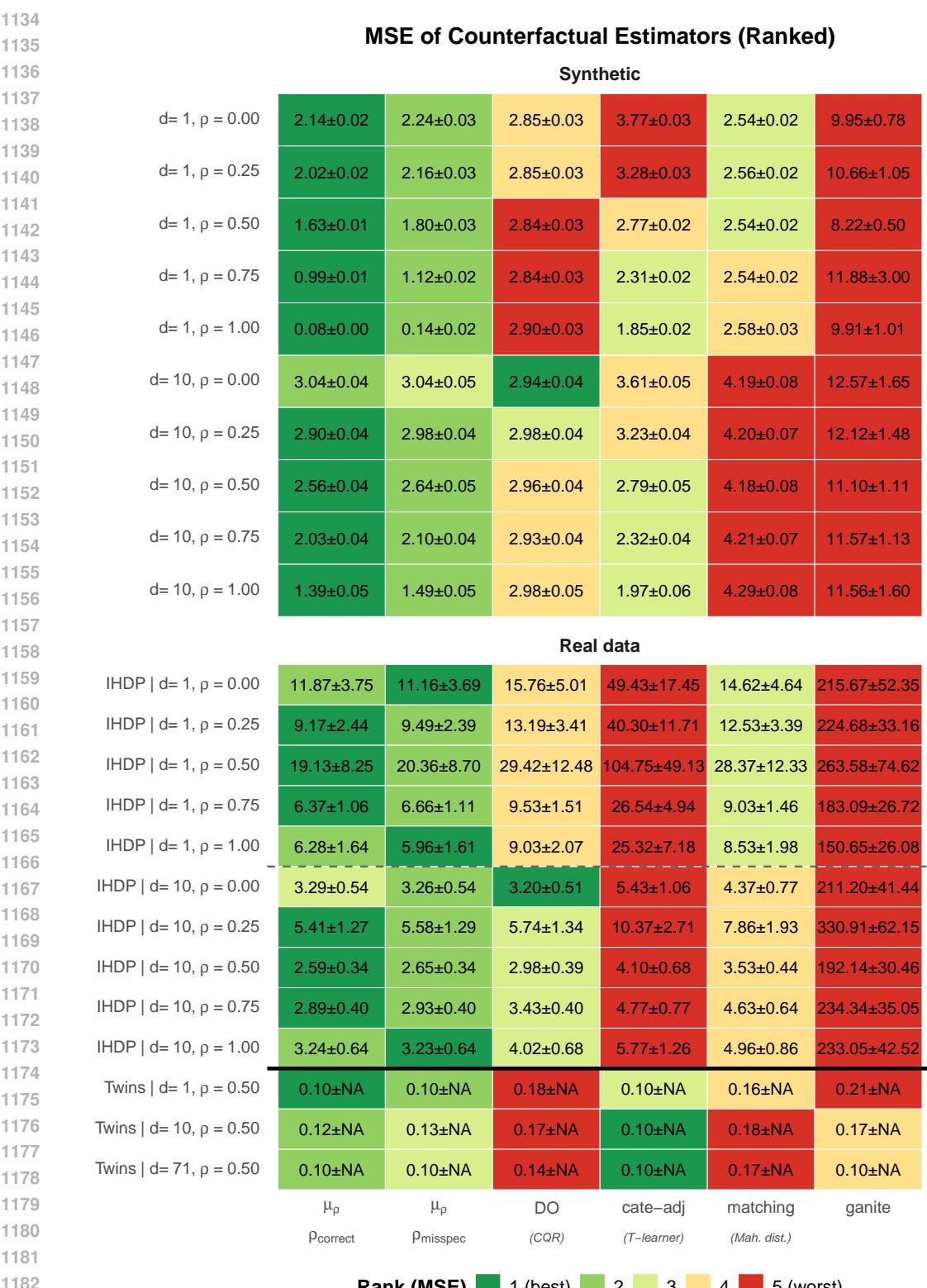

Figure 7: Extended version of Figure 2, additionally displaying the standard deviations of the MSE estimates within each cell.

## D  PROOFS

**Theorem 1** (Motivation and optimality under a perfect (asymptotic) scenario). *Let $x \in \mathcal{X}$, and $\rho = \mathrm{cor}\big(Y(0), Y(1) \mid X = x\big) \in [-1, 1]$. Assume a perfect scenario: $\big(Y(1), Y(0)\big) \mid X = x$ is Gaussian, $\hat{\mu}_t(x) = \mu_t(x)$ and suppose that we found conditionally valid prediction intervals:*

$$\mathbb{P}\big(Y(t) \leq \hat{\mu}_t(x) + u_t(x) \mid X = x\big) = 0.95, \quad \mathbb{P}\big(Y(t) \geq \hat{\mu}_t(x) - l_t(x) \mid X = x\big) = 0.95, \quad t = 0, 1.$$

*Then, $C_\rho$ prediction intervals from Definition 2 are optimal in a sense that it is the smallest set satisfying:*

$$\mathbb{P}\big(Y(1) \in C_\rho(X, Y(0)) \mid X = x, Y(0) = y\big) \geq 0.9,$$

*for any $y \in \mathbb{R}$. Moreover, $\hat{\mu}_\rho(x, y)$ is the optimal point predictor in the sense that it minimizes the mean squared error:*

$$\hat{\mu}_\rho(x, y) = \underset{c \in \mathbb{R}}{\mathrm{argmin}} \, \mathbb{E}\big[(Y(1) - c)^2 \mid X = x, Y(0) = y\big].$$

*Proof.* We use the following fact:

For a bivariate Gaussian random variables $(Z_1, Z_0)$:

$$\begin{pmatrix} Z_0 \\ Z_1 \end{pmatrix} \sim \mathcal{N}\left( \begin{pmatrix} \mu_0 \\ \mu_1 \end{pmatrix}, \begin{pmatrix} \sigma_0^2 & \rho\sigma_0\sigma_1 \\ \rho\sigma_0\sigma_1 & \sigma_1^2 \end{pmatrix} \right),$$

it is well known that:

$$Z_1 \mid Z_0 = z \sim \mathcal{N}\left( \mu_1 + \rho\frac{\sigma_1}{\sigma_0}(z - \mu_0), \; \sigma_1^2(1 - \rho^2) \right).$$

Moreover, the shortest prediction interval with a given coverage is symmetric around the mean.

First, we introduce some notation:

- Let $c := \Phi^{-1}(0.95) \approx 1.6449$ denote the 0.95 quantile of a standard Gaussian random variable.

- Let $\sigma_t^2(x) := \mathrm{Var}(Y(t) \mid X = x)$ denote the conditional variance.

- $\mu_t(x) + u_t(x) = \mathrm{Quantile}_{0.95}(Y(t) \mid X = x)$.

- Since $Y(t) \mid X = x$ is symmetrical around the mean, we have $l_t(x) = u_t(x)$. Therefore, $u_t(x) = c \cdot \sigma_t(x)$, by the standard form of the quantile function for a Gaussian distribution. Therefore, $\lambda(x) = \frac{\sigma_1(x)}{\sigma_0(x)}$.

Due to Gaussianity assumption, it holds that:

$$Y(1) \mid Y(0) = y, X = x \sim \mathcal{N}\left( \mu_1(x) + \rho\frac{\sigma_1(x)}{\sigma_0(x)}(y - \mu_0(x)), \; (1 - \rho^2)\sigma_1^2(x) \right)$$

which directly gives us

$$\mathbb{P}(Y(1) \leq \mu_1(x) + \rho\frac{\sigma_1(x)}{\sigma_0(x)}(y_0 - \mu_0(x)) + \sqrt{1 - \rho^2} \cdot c \cdot \sigma_1(x) \mid X = x, Y(0) = y_0) = 0.95.$$

Using our notation and previously established results, we get

$$\mathbb{P}(Y(1) \leq \hat{\mu}_\rho(x, y_0) + \sqrt{1 - \rho^2} \cdot u_1(x) \mid X = x, Y(0) = y_0) = 0.95,$$

and analogously

$$\mathbb{P}(Y(1) \geq \hat{\mu}_\rho(x, y_0) - \sqrt{1 - \rho^2} \cdot l_1(x) \mid X = x, Y(0) = y_0) = 0.95.$$

Hence, we proved that

$$\mathbb{P}\big(Y(1) \in C_\rho(X, Y(0)) \mid X = x, Y(0) = y_0\big) = 0.9.$$

The fact that $C_\rho$ prediction interval is the smallest possible interval achieving the desired coverage follows directly from symmetry+continuity of Gaussian variable.

The fact that $\hat{\mu}_\rho(x, y)$ is the optimal point predictor follows directly since

$$\hat{\mu}_\rho(x, y) = \mathbb{E}\big[Y(1) \mid X = x, Y(0) = y\big].$$

$\square$

**Theorem 2.** *Let $x \in \mathcal{X}$ and suppose $\big(Y(1), Y(0)\big) \mid X = x$ is Gaussian with $\rho = \mathrm{cor}\big(Y(1), Y(0) \mid X = x\big) \in [-1, 1]$.*

*Let $\hat{\mu}_t(x)$ be consistent estimators of $\mu_t(x)$, and assume the prediction interval widths $l_t(x), u_t(x)$ are asymptotically conditionally valid, i.e.,*

$$\lim_{n \to \infty} \mathbb{P}\big(Y(t) \leq \hat{\mu}_t(x) + u_t(x) \mid X = x\big) = 0.95, \qquad \lim_{n \to \infty} \mathbb{P}\big(Y(t) \geq \hat{\mu}_t(x) - l_t(x) \mid X = x\big) = 0.95,$$

*for $t = 0, 1$. Then, for any fixed $y \in \mathbb{R}$:*

1. *$\hat{\mu}_\rho(x, y)$ is a consistent estimator of the conditional mean,*

$$\hat{\mu}_\rho(x, y) \xrightarrow{p} \mathbb{E}\big[Y(1) \mid X = x, Y(0) = y\big], \quad \text{as } n \to \infty.$$

2. *The $C_\rho$ prediction intervals achieve asymptotic conditional coverage,*

$$\lim_{n \to \infty} \mathbb{P}\big(Y(1) \in C_\rho(X, Y(0)) \mid X = x, Y(0) = y\big) = 0.9.$$

*Proof.* Under the Gaussian assumption, Theorem 1 implies

$$\mathbb{E}\big[Y(1) \mid X = x, Y(0) = y\big] = \mu_1(x) + \rho \frac{\sigma_1(x)}{\sigma_0(x)}\big(y - \mu_0(x)\big). \qquad (8)$$

By consistency, $\hat{\mu}_t(x) \xrightarrow{p} \mu_t(x)$ for $t = 0, 1$. Moreover, since the upper and lower bounds converge to the 0.95 and 0.05 conditional quantiles of $Y(t) \mid X = x$, their total width satisfies

$$l_t(x) + u_t(x) \xrightarrow{p} \mathrm{Quantile}_{0.95}(Y(t) \mid X = x) - \mathrm{Quantile}_{0.05}(Y(t) \mid X = x) = 2z_{0.95}\sigma_t(x).$$

Thus,

$$\lambda(x) = \frac{l_1(x) + u_1(x)}{l_0(x) + u_0(x)} \xrightarrow{p} \frac{\sigma_1(x)}{\sigma_0(x)}.$$

Substituting into $\hat{\mu}_\rho(x, y)$,

$$\hat{\mu}_\rho(x, y) \xrightarrow{p} \mu_1(x) + \rho \frac{\sigma_1(x)}{\sigma_0(x)}\big(y - \mu_0(x)\big),$$

which coincides with equation 8, proving consistency of the point estimator.

For the prediction interval $C_\rho$, Theorem 1 further states that, under Gaussianity, $C_\rho(X, Y(0))$ is the minimal set achieving 90% conditional coverage for $Y(1) \mid X = x, Y(0) = y$. Since $l_t(x)$ and $u_t(x)$ converge to their true quantiles, the constructed interval converges to this optimal set. Hence,

$$\lim_{n \to \infty} \mathbb{P}\big(Y(1) \in C_\rho(X, Y(0)) \mid X = x, Y(0) = y\big) = 0.9.$$

$\square$

**Lemma 1** (Special cases of $\rho$). • *If $\rho = 0$ and $\tilde{C}_1(X)$ is marginally valid, then $C_\rho(X, Y(0))$ is also marginally valid:*

$$\mathbb{P}(Y(1) \in \tilde{C}_1(X)) \geq 0.9 \implies \mathbb{P}(Y(1) \in C_\rho(X, Y(0))) \geq 0.9.$$

*If additionally $Y(0) \perp\!\!\!\perp Y(1) \mid X = x$ and $\tilde{C}_1(X)$ is conditionally valid, then $C_\rho(X, Y(0))$ is also conditionally valid:*

$$\mathbb{P}(Y(1) \in \tilde{C}_1(X) \mid X = x) \geq 0.9 \implies \mathbb{P}(Y(1) \in C_\rho(X, Y(0)) \mid X = x, Y(0) = y) \geq 0.9,$$

*for any $x \in \mathcal{X}, y \in \mathcal{Y}$.*

- *If $\rho = \pm 1$ and $\mu(x, y_0) = \hat{\mu}(x, y_0)$, then*

$$\mathbb{P}(Y(1) \in C_\rho(X, Y(0)) \mid X = x, Y(0) = y) = 1.$$

*If we have confidence intervals satisfying $\mathbb{P}(\mu(x, y_0) \in \hat{\mu}(x, y_0) \pm r(x, y_0)) = 1 - \beta$, then*

$$\mathbb{P}(Y(1) \in C_\rho^{+CI}(X, Y(0)) \mid X = x, Y(0) = y) = 1 - \beta.$$

*Proof.* **Case $\rho = 0$:** By definition, $C_\rho(X, Y(0)) = \tilde{C}_1(X)$, so marginal validity is preserved. If $Y(0) \perp\!\!\!\perp Y(1) \mid X$, then conditioning on $Y(0)$ does not affect the validity, hence conditional validity also holds.

**Case $\rho = \pm 1$:** Perfect (anti-)correlation implies a deterministic linear relationship: for fixed $X = x$, we have

$$Y(1) = a_x + b_x Y(0) \quad \text{for some } a_x, b_x \in \mathbb{R}.$$

Thus,

$$\mathrm{Var}(Y(1) \mid X = x, Y(0) = y) = 0 \quad \Rightarrow \quad \mathbb{P}(Y(1) = \mu(x, y) \mid X = x, Y(0) = y) = 1.$$

If $\mu(x, y) = \hat{\mu}(x, y)$, then $C_\rho(x, y) = \{\mu(x, y)\}$, implying perfect coverage. If instead $\mu(x, y)$ lies in a confidence interval with coverage $1 - \beta$, then

$$\mathbb{P}(Y(1) \in C_\rho^{+CI}(x, y) \mid X = x, Y(0) = y) \geq 1 - \beta.$$

$\square$

