# OpenReview forum: "COUNTERFACTUAL PREDICTION WITH CROSS-WORLD DEPENDENCE"
_ICLR.cc/2026/Conference — ICLR 2026 Conference Withdrawn Submission_

### Official Review · Reviewer_FEmm · 2025-10-14

**Soundness:** 3
**Presentation:** 2
**Contribution:** 3
**Rating:** 2
**Confidence:** 3

**Summary:**

The paper introduces estimators that leverage cross-world dependency for a new causal quantity. The estimators include a point estimate and a prediction interval estimate. Some numerical evaluation results were shown about the proposed estimator.

**Strengths:**

The cross-world assumption is very interesting. Traditionally, the field of causal inference has treated Y(1) and Y(0) as two quantities that are "opposing" and "if one then not the other". The cross-world dependency assumption allows the identification of a new causal quantity that the paper studies, which was not accessible previously. This causal quantity is potentially of interest in many practical settings and might give rise to further theoretical interest in the community.

The paper seems to be among the first in this line of literature to tackle the above-mentioned problem.

The proposed methodology is simple, and the authors made substantial efforts to motivate the proposed methodology.

**Weaknesses:**

Assumption: Despite its novelty, I find the cross-world assumption to be very restrictive. Even though the author argued that in many real-world domains, experts may have some knowledge about $\rho(x)$, I believe it is unlikely that they can quantitatively trace out this function. Since this assumption is the central assumption of the paper, without proper justification, the contribution of this paper could be significantly undermined.

PS: I would suspect that this assumption must be verified in different applied domains: (1) whether or not domain experts can accurately identify $\rho(x)$, and (2) if the estimator's performance turns out to be superior.

Structure: The paper does not seem to have a part dedicated to describing related works in studying cross-world dependency. This part seems to be melded with the "preliminary" section, which could be confusing.

Notation inconsistency: In Definition 1, $\rho$ does not seem to be formally defined---does it represent the marginal version of $\rho(x)$?

Weak theories: It seems like there are no formal theoretical analyses of the properties of the proposed estimator. Additionally, the role of Theorem 1 "motivating" for Definition 1 seems vague to me.

Confusing experiment results: Some experiment results are counterintuitive. In Figure 2 "real data", why do some $\mu_\rho$ with misspecified $\rho$ perform even better than those with correctly specified $\rho$? For example, at IHDP $d = 1$ $\rho = 1$. The experiment results are not explained and analyzed.

**Questions:**

I would prefer hearing responses for the weaknesses identified above. Additionally:

Introduction part 1: Line 52, I am confused by the argument "this omission can lead to biased counterfactual predictions". To my knowledge, many estimators for CATE are consistent, so I am unsure what "biased" here refers to. Could the authors provide a simple math example?

Introduction part 2: Line 55 " Incorporating the factual outcome alongside the covariates can therefore refine individual-level predictions and improve the accuracy of estimated counterfactuals", this makes sense, but I did not find theorems in the paper supporting this argument. Could the authors elaborate on whether this theoretical result exists and/or where it is proved?

The authors should elaborate more on interpreting why Theorem 1 motivates Definition 1.

Could the authors also elaborate on why, even with a misspecified $\rho$, the proposed estimator outperforms other baselines?

---

> ### Author Response · Authors · 2025-11-17
>
> We thank Reviewer FEmm for an encouraging feedback. Below we answer your questions
>
> ### Weakness 1 (Assumption)
>
> We comment on this point in General Response Part of our review. We hope the paper’s transparency about this difficulty helps clarify the nature of the problem rather than undermining the contribution.
>
> ### Weakness 2 (Structure)
>
> We respectfully note that Section 2.1, titled \textit{Preliminaries About Cross-World Dependence}, is precisely intended to serve as the related-work and conceptual background for assumptions linking \(Y(0)\) and \(Y(1)\). To the best of our knowledge, there are no other papers modeling cross-world dependencies of this form beyond those already discussed there.
>
> ### Weakness 3 (notation inconsistency):
>
>  The reviewer is correct that we use \(\rho = 1\) as shorthand for \(\rho(x) = 1\) for all $x$. We have now added this clarification directly in Page 2 to avoid ambiguity.
>
> ### Weakness 4 (theory):
>
> Theorem 1 is intended as an asymptotic result showing that, under correct specification of the copula and as \(n \to \infty\), our estimator is optimal in the Gaussian setting. We do not claim finite-sample optimality, which is of course impossible without stronger assumptions. We have clarified this role of Theorem 1 and added explanatory text linking it more explicitly to Definition 1.
>
> ### Weakness 5 (Confusing results):
>
>  The cases where a misspecified $\rho$ appears to outperform the true $\rho$ are a consequence of Monte Carlo noise. Appendix B.2 reports the corresponding standard deviations, which are relatively large compared to the small differences observed in Figure 2. With 50 Monte Carlo repetitions, these fluctuations are expected. A larger number of repetitions (for example 500) would make the superiority of the correctly specified model consistently visible (but infeasible due to computational complexity of some methods). We have clarified this in the revision.
>
> ### Question 1 (Line 52, “bias”):
>
> This sentence was perhaps not well chosen. We meant that $E[Y(1) \mid X = x, Y(0) = y]$ and $E[Y(1) \mid X = x]$ can be very different. This has nothing to do with finite-sample bias in CATE estimation. We changed the sentence accordingly.
>
> ### Question 2 (Line 55, “refinement”).
>
>  The statement that conditioning on $Y(0)$ refines the prediction of $Y(1)$ follows directly from the well-known fact that conditioning always reduces mean squared error.  We have added a short explanation formalizing this argument.
>
> ### Question 3 (motivation of Theorem 1).
>
>  We have expanded the explanation following Theorem 1 to show that the optimal predictor in the Gaussian case depends linearly on the factual outcome, which motivates Definition 1 as the natural cross-world analog of a regression function.
>
> ### Question 4 (Why misspecification can still outperform baselines)
>
> Informal explanation: Even when \(\rho\) is slightly misspecified (say $0.25$ instead of $0.5$), the resulting error is still much smaller than the error produced by extreme choices such as $\rho=0$ or $\rho=1$ which are ``more misspecified''. This is obviously only true as long as our choice $\rho$ was not too horrible. This is more formally described by the Experiments in Section 4.3.
>
> Thank you for acknowledging the strength of the core ideas. Our aim is to contribute to a clearer understanding of what counterfactual inference can and cannot deliver, and we hope our explanations make this perspective more evident. We trust that our answers resolve the key concerns, and we remain happy to clarify anything further.

---

> > ### Comment · Reviewer_FEmm · 2025-11-19
> >
> > I appreciate the reviewer's time and effort in producing the response. There are some additional comments from my end.
> >
> > Weakness 2: There might have been a misunderstanding with my statement. I do recognize that the author has included a discussion of the related work, but I am suggesting that this part be a separate section named "related work" instead of "preliminary". The latter is often used for setting up notations and problem definitions.
> >
> > Weakness 4: Theorem 1 seems more like an identification result rather than an asymptotic result to me. For the latter, the consistency and convergence rate of the estimator have to be studied, which is not done in the paper. The author should note this distinction and clearly acknowledge it in the paper.
> >
> > Question 2: The justification "follows directly from the well-known fact that conditioning always reduces mean squared error" is too coarse to be a valid argument. This undermines the value of the paper, as one could argue that simply adding any additional covariates can achieve the same purpose. A more helpful way to analyze is to show how the estimation variance decreases as rho increases, meaning that Y(1) could be a helpful explanatory variable of Y(0) for prediction.
> >
> > However, to add to the previous point, since the authors acknowledged the erroneous claim pointed out by Question 1, I find the objective E[Y(1)|X=x, Y(0)=y] no longer well-motivated. The only justifying statement in the paper centers around refining the accuracy of the estimated counterfactuals at the individual level, which is arguably trivial. I suggest the author consider reframing how this objective can be better motivated in the introduction section.

---

> ### Author Response · Authors · 2025-11-21
>
> We thank the reviewer for the useful comments and discussion.  Below, we try to clarify the raised points:
>
> ### Weakness 2
>
> We apologize for misunderstanding your point; we changed the name of the subsection to Related work instead of preliminaries
>
> ### Weakness 4
>
> We explicitly added a new theorem about the consistency result:
>
> Let $x\in\mathcal{X}$ and suppose $\big(Y(1),Y(0)\big)\mid X=x $ is Gaussian with $\rho = \mathrm{cor}\big(Y(1),Y(0)\mid X=x\big)\in[-1,1]$.
>
> Let $\hat{\mu}(x)$ be consistent estimators of $\mu(x)$ (holds automatically for random forests for example), and assume the prediction interval widths $l_t(x),u_t(x)$ are asymptotically conditionally valid (holds automatically for CQR). Then, for any fixed $y\in\mathbb{R}$: $\hat{\mu}{\rho}(x,y)$ is a consistent estimator of the conditional mean,
> $$
> \hat{\mu}{\rho}(x,y) \to \mathbb{E}\big[Y(1)\mid X=x,Y(0)=y\big], \quad \text{as} \,n\to\infty.
> $$
>  The $C\rho$ prediction intervals achieve asymptotic conditional coverage,
> $$
> \lim_{n\to\infty}\mathbb{P}\big(Y(1)\in C{\rho}(X,Y(0))\mid X=x,Y(0)=y\big)=0.9.
> $$
>
>
> ### Question 2
>
> We fully agree with the reviewer that adding any additional covariate can improve prediction accuracy (equivalently, decrease variance). In our setting, the factual outcome $Y(0)$ indeed plays the role of an additional covariate. In this sense, it is trivial that using $Y(0)$ improves the accuracy, just as any additional covariate can improve the accuracy.  What is **not** trivial, however, is that unlike ordinary covariates, $Y(0)$ can only be used if we specify how $Y(0)$ and $Y(1)$ relate to each other, since unlike other covariates, they are never observed jointly. Does this answer your question?

---

> > ### Comment · Reviewer_FEmm · 2025-11-23
> >
> > Question 2: I appreciate the clarification. Since the author acknowledged $Y(0)$ can be viewed as an additional covariate, then wouldn't it be easier in practice to estimate $\mathbb{E}[Y(1)|X=x]$ by augmenting some additional useful covariates, than estimating $\mathbb{E}[Y(1)|X=x, Y(0)=y]$? From an applied practitioner's perspective, there seems to be no reason to opt for the latter to achieve the same purpose that can be done via the former. This question is much more fundamental and must be answered before even thinking about how to identify $\mathbb{E}[Y(1)|X=x, Y(0)=y]$. Without proper justification, the value of the paper will be significantly undermined.

---

> > > ### Author Response · Authors · 2025-11-25
> > > **Why model $E[Y(1)\mid X,Y(0)]$ instead of  $E[Y(1)\mid X]$ with more measured covariates?**
> > >
> > > We thank the reviewer for this important point. We agree that $Y(0)$ can be viewed as an additional covariate and that enlarging the covariate set in $E[Y(1)\mid X]$ may improve prediction performance. However, in applied settings, obtaining extra covariates $X$ is often expensive or entirely infeasible. In contrast, the factual outcome $Y(0)$ is already observed at no additional cost and can be directly used as a covariate.
> > >
> > > Importantly, $Y(0)$ is not simply another feature. It is often the most informative signal about an individual’s potential outcome under treatment, since it reflects the actual realized response for that individual, including latent characteristics that are generally unmeasurable. As a simple illustration, if the treatment has no effect, then $Y(0)=Y(1)$ and the counterfactual outcome can be predicted perfectly using $Y(0)$. Such perfect prediction is not possible using only $E[Y(1)\mid X]$, even with thousands of covariates measured.
> > >
> > > In this sense, the estimand $E[Y(1)\mid X,Y(0)]$ is not merely a reformulation of $E[Y(1)\mid X]$. It formalizes how to exploit the factual outcome as a covariate in settings where the two potential outcomes cannot be observed together. Without modeling cross-world dependence, $Y(0)$ cannot be incorporated in a principled way, even though it is free to observe, highly informative, and often more useful than collecting additional covariate data. Our work therefore identifies a meaningful target quantity that enables the use of $Y(0)$ for counterfactual inference rather than proposing a replacement for existing approaches based solely on $E[Y(1)\mid X]$. That said, the approach comes with technical challenges, such as specifying the dependence structure between the two potential outcomes.

---

### Official Review · Reviewer_qWZZ · 2025-10-25

**Soundness:** 3
**Presentation:** 2
**Contribution:** 3
**Rating:** 6
**Confidence:** 4

**Summary:**

This paper tackles the counterfactual prediction problem by making explicit a cross-world correlation parameter $\rho(x)$ to interpolate between ignoring the factual outcome ($\rho = 0$) and perfect dependence ($\rho = 1$). The authors propose a framework for point estimation and prediction intervals for counterfactual outcomes under a prespecified $\rho(x)$, deriving optimality under Gaussian assumptions, and extensive empirical evidence across synthetic, semi-synthetic, and real datasets.

**Strengths:**

1. The paper introduces an explicit model for the cross-world correlation ($\rho$), a factor largely overlooked in prior literature, and leverages it to develop novel methods for both point and interval estimation of counterfactual outcomes.
2. The empirical evaluation is comprehensive, spanning synthetic, semi-synthetic, and real-world datasets. The proposed method consistently outperforms existing baselines across these settings.
3. The authors conduct a thorough investigation into the model's robustness, including sensitivity to the misspecification of $\rho$ and performance in non-Gaussian settings, which strengthens the paper's empirical claims.

**Weaknesses:**

1. The proposed method's performance is highly dependent on the pre-specified correlation parameter $\rho(x)$, which requires prior knowledge that is often unavailable in practice. As Figure 3 demonstrates, performance degrades substantially under misspecification of $\rho$.
2. The theoretical guarantees for optimality are derived under a Gaussian assumption, which may not hold in many real-world applications and thus could limit the method's applicability. While empirical results suggest robustness in some non-Gaussian settings, these observations lack rigorous theoretical backing.
3. Key details regarding the conformal prediction methodology are relegated to the appendix. This makes it difficult for readers to fully grasp the approach for constructing prediction intervals without consulting supplementary material.
4. The authors state that most baselines implicitly assume $\rho=0$ or $\rho=1$. While Figure 2 shows the impact of a randomly misspecified $\rho$, the paper would be strengthened by reporting the performance of the proposed method when $\rho$ is fixed to 0 and 1. This would serve as an ablation study, helping to disentangle the performance gains attributable to the core modeling framework from the gains of using a well-specified $\rho$. Such an analysis would provide a fairer comparison to baselines and more decisively highlight the importance of the correlation parameter itself.

**Questions:**

1. Is there any theoretical guarantee that the optimality results of Theorem 1 can be generalized to non-Gaussian distributions?
2. The choice of $\rho=0.5$ for the Twins dataset requires clarification. While the ground-truth $\rho$ can be controlled in synthetic/semi-synthetic settings (e.g., Synthetic, IHDP), it is unknown for real-world datasets like Twins. Could the authors please clarify the rationale behind selecting $\rho=0.5$? Is this an assumption, or is it motivated by specific domain knowledge about this dataset?

---

> ### Author Response · Authors · 2025-11-17
>
> We thank Reviewer qWZZ for their thorough and positive review. Below we answer your questions:
>
> ### Question 1: Is there any theoretical guarantee that the optimality results of Theorem 1 can be generalized to non-Gaussian distributions?
>
> The main obstacle is conceptual rather than technical.  A consistent point estimate requires specifying the entire copula linking \(Y(0)\) and \(Y(1)\). The Gaussian copula is the most natural and interpretable choice, which is why we use it in Theorem 1. If one supplies any other copula, an analogue of Theorem 1 can be easily derived.
>
> The dependence structure is not identifiable from observed data, so any method that produces a single point estimate must fix a full joint distribution. In many applications there is no principled way to choose a non Gaussian copula, and only a few specialized domains offer meaningful alternatives (maybe Gumbel copula in some climate settings?). This is therefore an intrinsic feature of counterfactual prediction rather than a limitation of our approach. Without domain knowledge to guide the choice of copula, moving beyond the Gaussian case is not practically feasible.
>
>
>
> ### Question 2: The choice of $\rho=0.5$ for the Twins dataset requires clarification.
>
> Thank you for the comment, we appologise that this was not clearly explained in the main text. The choice was taken directly from Section 2.3.2 of Bodik et al. (2025), where the same dataset was discussed. In the revision, we have added this explanation into the main text to make the rationale clear.
>
> Regarding points 3 and 4, we agree that these are important suggestions. In the revision, we have moved two explanatory paragraphs on conformal inference from the appendix into the main text for improved clarity. We have also added two additional simulations with $\rho=0,1$, which serve as ablation studies and clarify the behavior of our method at the endpoints.
>
>
> ### Weakness 1 and 2 are answered in the General Response Part.
>
> ### Weakness 4
>
> This is a very good idea! Below are the results for one of the datasets (IHDP dataset with true ρ = 0.5):
>
> | Method                 | MSE (mean ± se) | Coverage (mean ± se) |
> | ---------------------- | --------------- | -------------------- |
> | \rho=\text{true} | 6.32 ± 1.04     | 0.898 ± 0.005        |
> | \rho=\text{misspec} | 6.76 ± 1.09     | 0.733 ± 0.037        |
> |  \rho=0          | 7.60 ± 1.21     | 0.919 ± 0.003        |
> |  \rho=1           | 7.48 ± 1.19     | 0.348 ± 0.007        |
>
> On this dataset, it seems that the improvement is around 20% in MSE. We will add such a table for all datasets (runtime is a few days).
>
> We hope these clarifications address your concerns and make the scope and limitations of our approach more transparent. We thank you again for the thoughtful feedback, which helped us improve the clarity and presentation of the paper.

---

### Official Review · Reviewer_neCD · 2025-10-31

**Soundness:** 1
**Presentation:** 2
**Contribution:** 1
**Rating:** 2
**Confidence:** 4

**Summary:**

This paper considers predicting counterfactual outcomes $E[Y(1) | X = x, Y(0) = y]$ by conditioning on both covariates and the untreated potential outcome. In general, since this quantity depends on the joint distribution (Y(0), Y(1)), it is not identified. So the paper's key contribution is to make explicit the role of "cross-world" correlation --- that is, specific a bound on $\rho(x) = Corr(Y(0), Y(1) \mid X = x)$. The paper proposes point estimators and prediction intervals for this quantity (see their discussion in Section 3 for further details). The authors show that under Gaussianity and other idealized conditions, the prediction intervals are optimal and their point estimators minimize MSE (Theorem 1). The authors illustrate their methods in experiments on synthetic datasets and two classic causal inference datasets.

**Strengths:**

The paper's two key ideas are potentially nice contributions. I see these two ideas as: (1) focusing on prediction intervals that condition on the factual outcome; and (2) introduce cross-world restrictions using the interpretable parameter $\rho(x)$ (as opposed to other rank invariance or copula based assumptions).

On (1): to my knowledge, this is the first work to systematically study prediction intervals of this form. Most counterfactual uncertainty quantification conditions only on $X$. Conditioning on the factual outcome Y(0) is conceptually natural—after observing that a patient remained healthy under control, we should tighten our uncertainty about their treated outcome.
On (2): formulating cross-world restrictions in terms of $\rho(x)$ is also natural and simple. It invites domain expertise as researchers might be able to reason about the choice of $\rho$ and lends itself naturally to sensitivity analyses.

**Weaknesses:**

I found that this paper overclaims what its theoretical contributions actually deliver.

a) The paper prominently frames its contribution around conditional coverage: that is, $\mathbb{P}(Y(1) \in C_\rho(x,y) \mid X = x, Y(0) = y) \geq 1 - \alpha$. However, as Barber et al. (2020)---which the authors themselves cite---establishes, conditional coverage is generally impossible in finite samples without very strong assumptions. Moreover: (i) the experiments rely on CQR, which only delivers marginal coverage; (ii) their own theoretical analysis shows that their interval inherits conditional validity if the baseline has it --- but CQR does not have it. So why lead with a conditional guarantee that cannot be satisfied in practice? The paper should be reframed entirely around marginal coverage guarantees, which are actually achievable (or suitable modifications of conditional coverage in the spirit of Gibbs et al. 2023). The current presentation misleads readers about what the method delivers.

b) Theorem 1 claims the $C_\rho$ intervals are "optimal" (smallest valid sets) and $\hat{\mu}_\rho$ minimizes MSE. However, this holds only under a ``perfect asymptotic scenario'' requiring: (i) Gaussianity; (ii) Oracle nuisances; (iii) conditionally valid baselines. It is not clear when this result would be relevant. (i) is unrealistic; (ii) does not apply in finite samples where estimation errors matter; and (iii) cannot generally be satisfied. More broadly, the paper does not provide an actual finite sample coverage theorem, which is the standard for conformal inference papers.

c) The paper claims early on that "Given $\rho$, we develop a consistent estimator and valid prediction intervals." As written, this suggests that specifying only the correlation (plus the marginals) is sufficient to identify $\mathbb{E}[Y(1) \mid X=x, Y(0)=y]$. This is not true in general since specifying only $\rho(x)$ does not pin down the full joint distribution of $(Y(0), Y(1)) \mid X=x$. Without additional assumptions (e.g., Gaussianity), the conditional mean $\mathbb{E}[Y(1) \mid X=x, Y(0)=y]$ is at best partially identified ---multiple joint distributions can share the same marginals and correlation but differ in their conditional expectations. But the paper is written as if this is more general. Furthermore, I don't see how the authors address the validity of the intervals if this quantity is partially identified.

d) The correlation $\rho(x)$ is also unidentifiable. But the paper provides very little guidance on how it could be reasonably chosen in practice. This is always the central challenge in sensitivity analysis frameworks -- is there a strategy for which this can be empirically calibrated? How might we elicit this information from practitioners? The paper would greatly benefit from expending more effort on how $\rho(x)$ might calibrated or elicited. The structural model was nice, but it only justifies $\rho \geq 0$.

**Questions:**

See my discussion of the paper's weaknesses.

---

> ### Author Response · Authors · 2025-11-17
>
> We thank Reviewer neCD for their thorough review and acknowledging the novelty of the paper. Below, we comment on your questions:
>
> ### Concern 1: we do not deliver conditional coverage guarantees
>
> We appreciate the reviewer’s concern. Our goal was not to claim we achieve conditional coverage in general, but to emphasize that our framework inherits whatever guarantees the underlying uncertainty quantification method provides. While CQR always provides marginal coverage, it \textbf{is} also conditionally valid in many practically relevant setting; for example, when covariates are discrete or if $n\to\infty$. Such scenarios arise frequently in applications with categorical features or when the sample size is huge. The broader conformal inference literature focuses on high-dimensional continuous settings precisely because they are challenging; we did not mean to suggest that our method overcomes these known impossibility results. We will adjust the framing accordingly, explicitly stating in the Section Contributions that we only provide marginal coverage as we only inherit the guarantees of the chosen UQ method.
>
> ### Concern 2: Assumptions of theorem 1 are too strong.
>
> We agree that the Gaussianity assumption is strong; we expand on that in the General Response Part. Regarding the other assumptions: in large samples $n\to\infty$, these conditions are satisfied by any standard nonparametric estimators like quantile random forests. Deriving finite-sample convergence rates for $\hat\mu_\rho$ is difficult because its accuracy depends on the rates of all the nuisance components. These rates are only available under very strong assumptions, which we explicitly chose not to impose.
>
> We view the asymptotic result in the same spirit as classical results such as the central limit theorem: it serves to clarify the structure of the idealized limit, even though the limit is never satisfied in finite-samples. We will revise the text to more clearly communicate the scope and interpretation of the theorem.
>
>
>
> ### Concern 3: The Gaussian-copula assumption is not sufficiently emphasized.
>
> We agree that this assumption must be absolutely explicit. We hoped that it was stressed enough, but perhaps we were mistaken. We added on several places very explicitly that we always work with Gaussian copula assumption.
>
> Is this strong assumption? Yes. But any method that attempts to produce a point estimate of a counterfactual inherently requires a full specification of the dependence structure; otherwise, the estimand is only partially identified. While more flexible copulas could be considered, they would require practitioners to supply even more detailed structural information. See our comment in the General Response Part.
>
>
> ### Concern 4: the paper provides very little guidance on how $\rho$ could be reasonably chosen in practice.
>
> We followed the guidance in Bodik et al. (2025), but we agree that more discussion would be helpful. We will expand the section on practitioner input and include additional examples from Bodik et al. (2025) illustrating plausible elicitation strategies and ranges. See also General Response Part.
>
>
> We appreciate your recognition that the main ideas of the paper are strong. We strongly believe this work represents a significant contribution to discussion about the limits of counterfactual inference and hope we can persuade you to support it. We hope our responses have fully addressed your questions and would also be delighted to continue our discussion and provide any further clarifications.

---

### Official Review · Reviewer_26pX · 2025-10-31

**Soundness:** 2
**Presentation:** 2
**Contribution:** 2
**Rating:** 2
**Confidence:** 4

**Summary:**

The paper tackles individual counterfactual prediction under cross-world dependence: i.e., conditional correlation between potential outcomes. They propose a consistent point estimator and intervals with nominal coverage. They argue that most existing approaches correspond to the extremes $\rho=0$ (ignore the factual outcome) or $\rho=1$ (constant effects), while their method interpolates between these and is theoretically optimal. 	I find the approach promising, although I am not persuaded that the “cross-world dependence” assumption adds clear value beyond existing methods, and have several questions.

**Strengths:**

- Makes the cross-world link explicit, and introduces not only a consistent estimator but also prediction intervals.
- Creates a clear knob for sensitivity analysis and a place to inject domain knowledge.
- Unifies existing approaches and supplies theoretical guarantees (consistency/optimality under stated conditions) .
- Compatible with off-the-shelf single-world predictors.

**Weaknesses:**

1. My primary hesitation is that the paper does not yet motivate or position itself clearly relative to recent advances in counterfactual prediction, which the current manuscript is entirely ignoring. In particular, there is a growing line of work that estimates counterfactual outcomes (or closely related conditional effects) under the potential–outcome framework, including:
- Kim, K., Kennedy, E., & Zubizarreta, J. (2022). Doubly robust counterfactual classification. Advances in Neural Information Processing Systems, 35, 34831-34845.
- McClean, A., Branson, Z., & Kennedy, E. H. (2024). Nonparametric estimation of conditional incremental effects. Journal of Causal Inference, 12(1), 20230024.
- Kim, K. (2025). Semiparametric Counterfactual Regression. arXiv preprint arXiv:2504.02694.
Importantly, these methods provide semiparametric efficiency (or efficiency-competitive rates) for their estimation/inferential procedures under weaker nonparametric conditions and also offer principled ways to interpolate potential outcomes via stochastic interventions.  I understand your target causal effects are different from those, but the paper needs to articulate when and why this particular alternative is preferable in practice.

2. The contribution hinges on the correlation between potential outcomes, which cannot be learned from observed data. Without a credible way to estimate or even bound the dependence between the two potential outcomes, the method just reduces to a sensitivity analysis rather than a learnable model. Any misspecification of this dependence directly translates into bias and miscalibrated uncertainty estimates, undermining the main inferential claims. Moreover, in my opinion, the assumption itself is difficult to justify or verify in realistic applications;  I believe domain experts rarely have concrete knowledge about cross-world dependence. Without stronger theoretical or empirical grounding for this assumption, the practical reliability and interpretability of the proposed framework remain limited.

3. The theoretical results rely on Gaussian behavior and near-oracle predictors, and are clean only in special cases (e.g., independence or perfect correlation). In the realistic middle regime, broad identification and coverage guarantees are missing.

4. The setup predicts an unobserved outcome using the observed outcome for the same unit—useful after the fact, but not for targeting or policy choices made beforehand. I don’t see clear descriptive or prescriptive benefits for decision makers.

**Questions:**

Please respond to the criticisms above with specific arguments and supporting evidence.

---

> ### Author Response · Authors · 2025-11-17
>
> We thank Reviewer 26pX for their review and for the thoughtful comments provided. Below we respond to the criticism.
>
> ### Concern 1: The paper ignores recent work on counterfactuals
>
> We appreciate the reviewer’s remarks. Existing studies (including the proposed papers) address prospective counterfactuals, which estimate potential outcomes prior to observing the factual one and form the basis of policy evaluation (Pearl’s second ladder of causation). Our focus is on retrospective counterfactuals, which aim to estimate what would have happened given what actually happened (third ladder).
>
> Retrospective counterfactuals address questions of causal responsibility rather than policy improvement. In law or insurance, one asks whether a specific harm would have occurred under an alternative action: questions fundamentally about what would have been, not what should be done, important for assessing individual responsibility. Prospective counterfactuals are perhaps more important in practice, but retrospective are still useful in some applications. This distinction motivates our methodological focus, which is very different from prospective estimators like the CATE-based ones. Therefore, the suggested papers (and most of the existing counterfactual literature) are not very relevant besides sharing a similar name. We recognize that our paper did not make this distinction clear enough in the introduction; we have clarified it in the revision by adding two more paragraphs about prospective vs retrospective counterfactual literature distinction.
>
> ### Concerns 2-4
> These concerns are answered in the General Response Part.
>
>
> We appreciate your recognition of the paper’s strength and hope our responses have fully addressed your questions. We strongly believe
> this work represents a significant contribution to discussion about the limits of counterfactual inference and hope we can persuade you to support it. We would also be delighted to continue our discussion and provide any further clarifications.

---

### Author Response · Authors · 2025-11-17
**General Response on two main concerns**

We thank all reviewers for their thoughtful feedback. Two concerns appeared across all reviews and they are the primary reasons behind the lower ratings. We address them here.

###  Concern 1: There is no credible way to estimate $\rho$, and without proper justification, the contribution of this paper could be significantly undermined.

We agree with this concern that we also acknowledge repeatedly in the paper. $\rho$ cannot be recovered from observed data and must be supplied externally. This limitation is fundamental: any method that outputs a point estimate of a counterfactual (not merely a partial-identification set) must specify the cross-world dependence. Our goal is not to bypass this impossibility but to make it explicit and transparent. In this sense, our contribution is conceptual, not methodological: we show precisely what must be assumed and where identifiability ends, rather than implicitly assuming independence. Since this limitation is inherent to the problem and cannot be resolved by any data driven method, it should not be viewed as a gap in our methodology.

###  Concern 2: The assumption of Gaussianity is strong and very limiting.

We also agree that Gaussianity is restrictive; this relates to Concern 1. When the variables are non-Gaussian, the correlation alone does not fully specify the copula, and additional information about the joint dependence would be required. The assumption could be relaxed to any fixed copula family, and our results would translate directly. However, doing so requires even more detailed practitioner input about the shape of dependence, which in many domains is even harder to elicit than $\rho$  itself. In our experiments, the bias arising from misspecifying the copula family appears to be small relative to the overall uncertainty, so we focused only on Gaussian case for interpretability. We will revise the manuscript to clarify this role and the trade-offs involved.

###  Concern 3: Lack of consistency results

We now include additional formal consistency theorem in the paper:


Theorem:
Let $x\in\mathcal{X}$ and suppose $\big(Y(1),Y(0)\big)\mid X=x $ is Gaussian with $\rho(x) = \mathrm{cor}\big(Y(1),Y(0)\mid X=x\big)\in[-1,1]$.

Let $\hat{\mu}(x)$ be consistent estimators of $\mu(x)$, and assume the prediction interval widths $l(x),u(x)$ are asymptotically conditionally valid (Note: This holds for many nonparametric estimators under mild smoothness assumptions, including random forests for estimating $\hat{\mu}(x)$ and CQR using quantile random forests for prediction intervals).

Then, for any fixed $y\in\mathbb{R}$: $\hat{\mu}\rho(x,y)$ is a consistent estimator of the conditional mean,
$$
\hat{\mu}\rho(x,y)\xrightarrow{p} \mathbb{E}\big[Y(1)\mid X=x,Y(0)=y\big], \quad \text{as} \,n\to\infty.
$$
The $C\rho$ prediction intervals achieve asymptotic conditional coverage,
$$
    \lim_{n\to\infty}\mathbb{P}\big(Y(1)\in C\rho(X,Y(0))\mid X=x,Y(0)=y\big)=0.9.
$$
Moreover,  $C\rho$ prediction intervals are asymptotically optimal in a sense that it is the smallest set satisfying Equation 3 as $n\to\infty$.


Although this result follows directly from Theorem 1, we agree that this was not immediately clear from the original presentation, and we have revised the text accordingly for clarity.

---

### Note · Authors · 2025-12-03

I have read and agree with the venue's withdrawal policy on behalf of myself and my co-authors.